# EFFICIENTLY TESTING LOCAL OPTIMALITY AND ESCAPING SADDLES FOR RELU NETWORKS

**Chulhee Yun, Suvrit Sra & Ali Jadbabaie**
Massachusetts Institute of Technology
Cambridge, MA 02139, USA
{`chulheey`,`suvrit`,`jadbabai`}@mit.edu

## ABSTRACT

We provide a theoretical algorithm for checking local optimality and escaping saddles at *nondifferentiable* points of empirical risks of two-layer ReLU networks. Our algorithm receives any parameter value and returns: *local minimum*, *second-order stationary point*, or *a strict descent direction*. The presence of $M$ data points on the nondifferentiability of the ReLU divides the parameter space into at most $2^M$ regions, which makes analysis difficult. By exploiting polyhedral geometry, we reduce the total computation down to one convex quadratic program (QP) for each hidden node, $O(M)$ (in)equality tests, and one (or a few) nonconvex QP. For the last QP, we show that our specific problem can be solved efficiently, in spite of nonconvexity. In the benign case, we solve one equality constrained QP, and we prove that projected gradient descent solves it exponentially fast. In the bad case, we have to solve a few more inequality constrained QPs, but we prove that the time complexity is exponential only in the number of inequality constraints. Our experiments show that either benign case or bad case with very few inequality constraints occurs, implying that our algorithm is efficient in most cases.

## 1 INTRODUCTION

Empirical success of deep neural networks has sparked great interest in the theory of deep models. From an optimization viewpoint, the biggest mystery is that deep neural networks are successfully trained by gradient-based algorithms despite their nonconvexity. On the other hand, it has been known that training neural networks to global optimality is NP-hard (Blum & Rivest, 1988). It is also known that even checking *local optimality* of nonconvex problems can be NP-hard (Murty & Kabadi, 1987). Bridging this gap between theory and practice is a very active area of research, and there have been many attempts to understand why optimization works well for neural networks, by studying the loss surface (Baldi & Hornik, 1989; Yu & Chen, 1995; Kawaguchi, 2016; Soudry & Carmon, 2016; Nguyen & Hein, 2017; 2018; Safran & Shamir, 2018; Laurent & Brecht, 2018; Yun et al., 2019; 2018; Zhou & Liang, 2018; Wu et al., 2018; Shamir, 2018) and the role of (stochastic) gradient-based methods (Tian, 2017; Brutzkus & Globerson, 2017; Zhong et al., 2017; Soltanolkotabi, 2017; Li & Yuan, 2017; Zhang et al., 2018; Brutzkus et al., 2018; Wang et al., 2018; Li & Liang, 2018; Du et al., 2018a;b;c; Allen-Zhu et al., 2018; Zou et al., 2018; Zhou et al., 2019).

One of the most important beneficial features of convex optimization is the existence of an optimality test (e.g., norm of the gradient is smaller than a certain threshold) for termination, which gives us a certificate of (approximate) optimality. In contrast, many practitioners in deep learning rely on running first-order methods for a fixed number of epochs, without good termination criteria for the optimization problem. This means that the solutions that we obtain at the end of training are not necessarily global or even local minima. Yun et al. (2018; 2019) showed efficient and simple global optimality tests for deep *linear* neural networks, but such optimality tests cannot be extended to general nonlinear neural networks, mainly due to nonlinearity in activation functions.

Besides nonlinearity, in case of ReLU networks significant additional challenges in the analysis arise due to nondifferentiability, and obtaining a precise understanding of the nondifferentiable points is still elusive. ReLU activation function $h(t) = \max\{t, 0\}$ is nondifferentiable at $t = 0$. This means that, for example, the function $f(w, b) := (h(w^T x + b) - 1)^2$ is nondifferentiable for any $(w, b)$ satisfying $w^T x + b = 0$. See Figure 1 for an illustration of how the empirical risk of a ReLU network

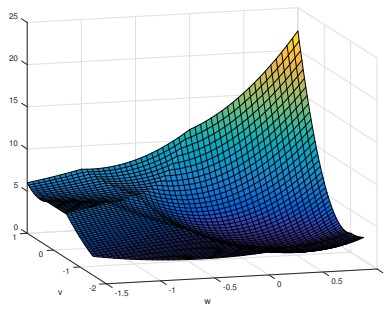

(a) A 3-d surface plot of $f(w, v)$.

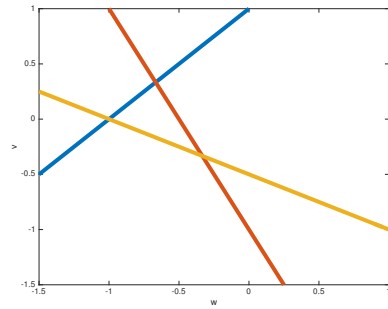

(b) Nondifferentiable points on $(w, v)$ plane.

Figure 1: An illustration of the empirical risk of a ReLU network. The plotted function is $f(w, v) := (h(w - v + 1) - 2)^2 + (h(2w + v + 1) - 1)^2 + (h(w + 2v + 1) - 0.5)^2$, where $h$ is the ReLU function. (a) A 3-d surface plot of the function. One can see that there are sharp ridges in the function. (b) A plot of nondifferentiable points on the $(w, v)$ plane. The blue line correspond to the line $w - v + 1 = 0$, the red to $2w + v + 1 = 0$, and the yellow to $w + 2v + 1 = 0$.

looks like. Although the plotted function does not exactly match the definition of empirical risk we study in this paper, the figures help us understand that the empirical risk is continuous but piecewise differentiable, with affine hyperplanes on which the function is nondifferentiable.

Such nondifferentiable points lie in a set of measure zero, so one may be tempted to overlook them as "non-generic." However, when studying critical points we cannot do so, as they are precisely such "non-generic" points. For example, Laurent & Brecht (2018) study one-hidden-layer ReLU networks with hinge loss and note that except for piecewise constant regions, local minima always occur on nonsmooth boundaries. Probably due to difficulty in analysis, there have not been other works that handle such nonsmooth points of losses and prove results that work for *all* points. Some theorems (Soudry & Carmon, 2016; Nguyen & Hein, 2018) hold "almost surely"; some assume differentiability or make statements only for differentiable points (Nguyen & Hein, 2017; Yun et al., 2019); others analyze population risk, in which case the nondifferentiability disappears after taking expectation (Tian, 2017; Brutzkus & Globerson, 2017; Du et al., 2018b; Safran & Shamir, 2018; Wu et al., 2018).

## 1.1 SUMMARY OF OUR RESULTS

In this paper, we take a step towards understanding nondifferentiable points of the empirical risk of one-hidden-layer ReLU(-like) networks. Specifically, we provide a theoretical algorithm that tests second-order stationarity for *any* point of the loss surface. It takes an input point and returns:

(a) The point is a local minimum; or
(b) The point is a second-order stationary point (SOSP); or
(c) A descent direction in which the function value strictly decreases.

Therefore, we can test whether a given point is a SOSP. If not, the test extracts a guaranteed direction of descent that helps continue minimization. With a proper numerical implementation of our algorithm (although we leave it for future work), one can run a first-order method until it gets stuck near a point, and run our algorithm to test for optimality/second-order stationarity. If the point is an SOSP, we can terminate without further computation over many epochs; if the point has a descent direction, our algorithm will return a descent direction and we can continue on optimizing. Note that the descent direction may come from the second-order information; our algorithm even allows us to escape nonsmooth second-order saddle points. This idea of mixing first and second-order methods has been explored in differentiable problems (see, for example, Carmon et al. (2016); Reddi et al. (2017) and references therein), but not for nondifferentiable ReLU networks.

The key computational challenge in constructing our algorithm for nondifferentiable points is posed by data points that causes input $0$ to the ReLU hidden node(s). Such data point bisects the parameter space into two halfspaces with different "slopes" of the loss surface, so one runs into nondifferen-

tiability. We define these data points to be **boundary data points**. For example, in Figure 1b, if the input to our algorithm is $(w, v) = (-2/3, 1/3)$, then there are two boundary data points: "blue" and "red." If there are $M$ such boundary data points, then in the worst case the parameter space divides into $2^M$ regions, or equivalently, there are $2^M$ "pieces" of the function that surround the input point. Of course, naively testing each region will be very inefficient; in our algorithm, we overcome this issue by a clever use of polyhedral geometry. Another challenge comes from the second-order test, which involves solving nonconvex QPs. Although QP is NP-hard in general (Pardalos & Vavasis, 1991), we prove that the QPs in our algorithm are still solved efficiently in most cases. We further describe the challenges and key ideas in Section 2.1.

**Notation.** For a vector $v$, $[v]_i$ denotes its $i$-th component, and $\|v\|_H := \sqrt{v^T H v}$ denotes a semi-norm where $H$ is a positive semidefinite matrix. Given a matrix $A$, we let $[A]_{i,j}$, $[A]_{i,\cdot}$, and $[A]_{\cdot,j}$ be $A$'s $(i, j)$-th entry, the $i$-th row, and the $j$-th column, respectively.

## 2 PROBLEM SETTING AND KEY IDEAS

We consider a one-hidden-layer neural network with input dimension $d_x$, hidden layer width $d_h$, and output dimension $d_y$. We are given $m$ pairs of data points and labels $(x_i, y_i)_{i=1}^m$, where $x_i \in \mathbb{R}^{d_x}$ and $y_i \in \mathbb{R}^{d_y}$. Given an input vector $x$, the output of the network is defined as $Y(x) := W_2 h(W_1 x + b_1) + b_2$, where $W_2 \in \mathbb{R}^{d_y \times d_h}$, $b_2 \in \mathbb{R}^{d_y}$, $W_1 \in \mathbb{R}^{d_h \times d_x}$, and $b_1 \in \mathbb{R}^{d_h}$ are the network parameters. The activation function $h$ is *"ReLU-like,"* meaning $h(t) := \max\{s_+ t, 0\} + \min\{s_- t, 0\}$, where $s_+ > 0, s_- \geq 0$ and $s_+ \neq s_-$. Note that ReLU and Leaky-ReLU are members of this class. In training neural networks, we are interested in minimizing the empirical risk

$$\mathfrak{R}((W_j, b_j)_{j=1}^2) = \sum\nolimits_{i=1}^m \ell(Y(x_i), y_i) = \sum\nolimits_{i=1}^m \ell(W_2 h(W_1 x_i + b_1) + b_2, y_i),$$

over the parameters $(W_j, b_j)_{j=1}^2$, where $\ell(w, y) : \mathbb{R}^{d_y} \times \mathbb{R}^{d_y} \mapsto \mathbb{R}$ is the loss function. We make the following assumptions on the loss function and the training dataset:

**Assumption 1.** *The loss function $\ell(w, y)$ is twice differentiable and convex in $w$.*

**Assumption 2.** *No $d_x + 1$ data points lie on the same affine hyperplane.*

Assumption 1 is satisfied by many standard loss functions such as squared error loss and cross-entropy loss. Assumption 2 means, if $d_x = 2$ for example, no three data points are on the same line. Since real-world datasets contain noise, this assumption is also quite mild.

### 2.1 CHALLENGES AND KEY IDEAS

In this section, we explain the difficulties at nondifferentiable points and ideas on overcoming them. Our algorithm is built from first principles, rather than advanced tools from nonsmooth analysis.

**Bisection by boundary data points.** Since the activation function $h$ is nondifferentiable at 0, the behavior of data points at the "boundary" is decisive. Consider a simple example $d_h = 1$, so $W_1$ is a row vector. If $W_1 x_i + b_1 \neq 0$, then the sign of $(W_1 + \Delta_1) x_i + (b_1 + \delta_1)$ for any small perturbations $\Delta_1$ and $\delta_1$ stays invariant. In contrast, when there is a point $x_i$ on the "boundary," i.e., $W_1 x_i + b_1 = 0$, then the slope depends on the direction of perturbation, leading to nondifferentiability. As mentioned earlier, we refer to such data points as **boundary data points**. When $\Delta_1 x_i + \delta_1 \geq 0$,

$$h((W_1 + \Delta_1) x_i + (b_1 + \delta_1)) = h(\Delta_1 x_i + \delta_1) = s_+(\Delta_1 x_i + \delta_1) = h(W_1 x_i + b_1) + s_+(\Delta_1 x_i + \delta_1),$$

and similarly, the slope is $s_-$ for $\Delta_1 x_i + \delta_1 \leq 0$. This means that the "gradient" (as well as higher order derivatives) of $\mathfrak{R}$ depends on *direction* of $(\Delta_1, \delta_1)$.

Thus, every boundary data point $x_i$ bisects the space of perturbations $(\Delta_j, \delta_j)_{j=1}^2$ into two halfspaces by introducing a hyperplane through the origin. The situation is even worse if we have $M$ boundary data points: they lead to a worst case of $2^M$ regions. Does it mean that we need to test all $2^M$ regions separately? We show that there is a way to get around this issue, but before that, we first describe how to test local minimality or stationarity for each region.

**Second-order local optimality conditions.** We can expand $\mathfrak{R}((W_j + \Delta_j, b_j + \delta_j)_{j=1}^2)$ and obtain the following Taylor-like expansion for small enough perturbations (see Lemma 2 for details)

$$\mathfrak{R}(z + \eta) = \mathfrak{R}(z) + g(z, \eta)^T \eta + \tfrac{1}{2} \eta^T H(z, \eta) \eta + o(\|\eta\|^2), \tag{1}$$

where $z$ is a vectorized version of all parameters $(W_j, b_j)_{j=1}^2$ and $\eta$ is the corresponding vector of perturbations $(\Delta_j, \delta_j)_{j=1}^2$. Notice now that in (1), at nondifferentiable points the usual Taylor expansion does not exist, but the corresponding "gradient" $g(\cdot)$ and "Hessian" $H(\cdot)$ now depend on the *direction* of perturbation $\eta$. Also, the space of $\eta$ is divided into at most $2^M$ regions, and $g(z, \eta)$ and $H(z, \eta)$ are piecewise-constant functions of $\eta$ whose "pieces" correspond to the regions. One could view this problem as $2^M$ constrained optimization problems and try to solve for KKT conditions at $z$; however, we provide an approach that is developed from first principles and solves all $2^M$ problems efficiently.

Given this expansion (1) and the observation that derivatives stay invariant with respect to scaling of $\eta$, one can note that (a) $g(z, \eta)^T \eta \geq 0$ for all $\eta$, and (b) $\eta^T H(z, \eta)\eta \geq 0$ for all $\eta$ such that $g(z, \eta)^T \eta = 0$ are necessary conditions for local optimality of $z$, thus $z$ is a "SOSP" (see Definition 2.2). The conditions become sufficient if (b) is replaced with $\eta^T H(z, \eta)\eta > 0$ for all $\eta \neq \mathbf{0}$ such that $g(z, \eta)^T \eta = 0$. In fact, this is a generalized version of second-order necessary (or sufficient) conditions, i.e., $\nabla f = \mathbf{0}$ and $\nabla^2 f \succeq \mathbf{0}$ (or $\nabla^2 f \succ \mathbf{0}$), for twice differentiable $f$.

**Efficiently testing SOSP for exponentially many regions.** Motivated from the second-order expansion (1) and necessary/sufficient conditions, our algorithm consists of three steps:

(a) Testing first-order stationarity (in the Clarke sense, see Definition 2.1),
(b) Testing $g(z, \eta)^T \eta \geq 0$ for all $\eta$,
(c) Testing $\eta^T H(z, \eta)\eta \geq 0$ for $\{\eta \mid g(z, \eta)^T \eta = 0\}$.

The tests are executed from Step (a) to (c). Whenever a test fails, we get a strict descent direction $\eta$, and the algorithm returns $\eta$ and terminates. Below, we briefly outline each step and discuss how we can efficiently perform the tests. We first check first-order stationarity because it makes Step (b) easier. Step (a) is done by solving one convex QP per each hidden node. For Step (b), we formulate linear programs (LPs) per each $2^M$ region, so that checking whether all LPs have minimum cost of zero is equivalent to checking $g(z, \eta)^T \eta \geq 0$ for all $\eta$. Here, the feasible sets of LPs are pointed polyhedral cones, whereby it suffices to check only the extreme rays of the cones. It turns out that there are only $2M$ extreme rays, each shared by $2^{M-1}$ cones, so testing $g(z, \eta)^T \eta \geq 0$ can be done with only $O(M)$ inequality/equality tests instead of solving exponentially many LPs. In Step (b), we also record the **flat extreme rays**, which are defined to be the extreme rays with $g(z, \eta)^T \eta = 0$, for later use in Step (c).

In Step (c), we test if the second-order perturbation $\eta^T H(\cdot)\eta$ can be negative, for directions where $g(z, \eta)^T \eta = 0$. Due to the constraint $g(z, \eta)^T \eta = 0$, the second-order test requires solving constrained nonconvex QPs. In case where there is no flat extreme ray, we need to solve only one equality constrained QP (ECQP). If there exist flat extreme rays, a few more inequality constrained QPs (ICQPs) are solved. Despite NP-hardness of general QPs (Pardalos & Vavasis, 1991), we prove that the specific form of QPs in our algorithm are still tractable in most cases. More specifically, we prove that projected gradient descent on ECQPs converges/diverges exponentially fast, and each step takes $O(p^2)$ time ($p$ is the number of parameters). In case of ICQPs, it takes $O(p^3 + L^3 2^L)$ time to solve the QP, where $L$ is the number of boundary data points that have flat extreme rays ($L \leq M$). Here, we can see that if $L$ is small enough, the ICQP can still be solved in polynomial time in $p$. At the end of the paper, we provide empirical evidences that the number of flat extreme rays is zero or very few, meaning that in most cases we can solve the QP efficiently.

## 2.2 PROBLEM-SPECIFIC NOTATION AND DEFINITION

In this section, we define a more precise notion of generalized stationary points and introduce some additional symbols that will be helpful in streamlining the description of our algorithm in Section 3. Since we are dealing with nondifferentiable points of nonconvex $\mathfrak{R}$, usual notions of (sub)gradients do not work anymore. Here, *Clarke subdifferential* is a useful generalization (Clarke et al., 2008):

**Definition 2.1** (FOSP, Theorem 6.2.5 of Borwein & Lewis (2010)). *Suppose that a function $f(z) : \Omega \mapsto \mathbb{R}$ is locally Lipschitz around the point $z^* \in \Omega$, and differentiable in $\Omega \setminus \mathcal{W}$ where $\mathcal{W}$ has Lebesgue measure zero. Then the Clarke differential of $f$ at $z^*$ is*

$$\partial_z f(z^*) := \text{cvxhull}\{\lim_k \nabla f(z_k) \mid z_k \to z^*, z_k \notin \mathcal{W}\}.$$

*If $\mathbf{0} \in \partial_z f(z^*)$, we say $z^*$ is a first-order stationary point (FOSP).*

From the definition, we can note that Clarke subdifferential $\partial_z \mathfrak{R}(z^*)$ is the convex hull of all the possible values of $g(z^*, \eta)$ in (1). For parameters $(W_j, b_j)_{j=1}^2$, let $\partial_{W_j} f(z^*)$ and $\partial_{b_j} f(z^*)$ be the Clarke differential w.r.t. to $W_j$ and $b_j$, respectively. They are the projection of $\partial_z f(z^*)$ onto the space of individual parameters. Whenever the point $z^*$ is clear (e.g. our algorithm), we will omit $(z^*)$ from $f(z^*)$. Next, we define second-order stationary points for the empirical risk $\mathfrak{R}$. Notice that this generalizes the definition of SOSP for differentiable functions $f$: $\nabla f = \mathbf{0}$ and $\nabla^2 f \succeq \mathbf{0}$.

**Definition 2.2** (SOSP). *We call $z^*$ is a second-order stationary point (SOSP) of $\mathfrak{R}$ if (1) $z^*$ is a FOSP, (2) $g(z^*, \eta)^T \eta \geq 0$ for all $\eta$, and (3) $\eta^T H(z^*, \eta)\eta \geq 0$ for all $\eta$ such that $g(z^*, \eta)^T \eta = 0$.*

Given an input data point $x \in \mathbb{R}^{d_x}$, we define $O(x) := h(W_1 x + b_1)$ to be the output of hidden layer. We note that the notation $O(\cdot)$ is overloaded with the big O notation, but their meaning will be clear from the context. Consider perturbing parameters $(W_j, b_j)_{j=1}^2$ with $(\Delta_j, \delta_j)_{j=1}^2$, then the perturbed output $\tilde{Y}(x)$ of the network and the amount of perturbation $dY(x)$ can be expressed as

$$dY(x) := \tilde{Y}(x) - Y(x) = \Delta_2 O(x) + \delta_2 + (W_2 + \Delta_2)J(x)(\Delta_1 x + \delta_1),$$

where $J(x)$ can be thought informally as the "Jacobian" matrix of the hidden layer. The matrix $J(x) \in \mathbb{R}^{d_h \times d_h}$ is diagonal, and its $k$-th diagonal entry is given by

$$[J(x)]_{k,k} := \begin{cases} h'([W_1 x + b_1]_k) & \text{if } [W_1 x + b_1]_k \neq 0 \\ h'([\Delta_1 x + \delta_1]_k) & \text{if } [W_1 x + b_1]_k = 0, \end{cases}$$

where $h'$ is the derivative of $h$. We define $h'(0) := s_+$, which is okay because it is always multiplied with zero in our algorithm. For boundary data points, $[J(x)]_{k,k}$ *depends* on the direction of perturbations $[\Delta_1 \ \delta_1]_{k,\cdot}$, as noted in Section 2.1. We additionally define $dY_1(x)$ and $dY_2(x)$ to separate the terms in $dY(x)$ that are linear in perturbations versus quadratic in perturbations.

$$dY_1(x) := \Delta_2 O(x) + \delta_2 + W_2 J(x)(\Delta_1 x + \delta_1), \ \ dY_2(x) := \Delta_2 J(x)(\Delta_1 x + \delta_1).$$

For simplicity of notation for the rest of the paper, we define for all $i \in [m] := \{1, \ldots, m\}$,

$$\bar{x}_i := \begin{bmatrix} x_i^T & 1 \end{bmatrix}^T \in \mathbb{R}^{d_x + 1}, \ \ \nabla \ell_i := \nabla_w \ell(Y(x_i), y_i), \ \ \nabla^2 \ell_i := \nabla_w^2 \ell(Y(x_i), y_i).$$

In our algorithm and its analysis, we need to give a special treatment to the boundary data points. To this end, for each node $k \in [d_h]$ in the hidden layer, define *boundary index set* $B_k$ as

$$B_k := \{i \in [m] \mid [W_1 x_i + b_1]_k = 0\}.$$

The subspace spanned by vectors $\bar{x}_i$ for in $i \in B_k$ plays an important role in our tests; so let us define a symbol for it, as well as the cardinality of $B_k$ and their sum:

$$\mathcal{V}_k := \text{span}\{\bar{x}_i \mid i \in B_k\}, \ \ M_k := |B_k|, \ \ M := \sum_{k=1}^{d_h} M_k.$$

For $k \in [d_h]$, let $v_k^T \in \mathbb{R}^{1 \times (d_x + 1)}$ be the $k$-th row of $[\Delta_1 \ \delta_1]$, and $u_k \in \mathbb{R}^{d_y}$ be the $k$-th column of $\Delta_2$. Next, we define the total number of parameters $p$, and vectorized perturbations $\eta \in \mathbb{R}^p$:

$$p := d_y + d_y d_h + d_h(d_x + 1), \ \ \eta^T := \begin{bmatrix} \delta_2^T & u_1^T & \cdots & u_{d_h}^T & v_1^T & \cdots & v_{d_h}^T \end{bmatrix}.$$

Also let $z \in \mathbb{R}^p$ be vectorized parameters $(W_j, b_j)_{j=1}^2$, packed in the same order as $\eta$.

Define a matrix $C_k := \sum_{i \notin B_k} h'([W_1 x_i + b_1]_k) \nabla \ell_i \bar{x}_i^T \in \mathbb{R}^{d_y \times (d_x + 1)}$. This quantity appears multiplie times and does not depend on the perturbation, so it is helpful to have a symbol for it.

We conclude this section by presenting one of the implications of Assumption 2 in the following lemma, which we will use later. The proof is simple, and is presented in Appendix B.1.

**Lemma 1.** *If Assumption 2 holds, then $M_k \leq d_x$ and the vectors $\{\bar{x}_i\}_{i \in B_k}$ are linearly independent.*

## 3 TEST ALGORITHM FOR SECOND-ORDER STATIONARITY

In this section, we present SOSP-CHECK in Algorithm 1, which takes an arbitrary tuple $(W_j, b_j)_{j=1}^2$ of parameters as input and checks whether it is a SOSP. We first present a lemma that shows the explicit form of the perturbed empirical risk $\mathfrak{R}(z + \eta)$ and identify first and second-order perturbations. The proof is deferred to Appendix B.2.

---

**Algorithm 1** SOSP-CHECK (Rough pseudocode)

Input: A tuple $(W_j, b_j)_{j=1}^2$ of $\mathfrak{R}(\cdot)$.
1: Test if $\partial_{W_2}\mathfrak{R} = \{\mathbf{0}_{d_y \times d_h}\}$ and $\partial_{b_2}\mathfrak{R} = \{\mathbf{0}_{d_y}\}$.
2: **for** $k \in [d_h]$ **do**
3:     **if** $M_k > 0$ **then**
4:         Test if $\mathbf{0}_{d_x+1}^T \in \partial_{[W_1\ b_1]_{k,\cdot}}\mathfrak{R}$.
5:         Test if $g_k(z, v_k)^T v_k \geq 0$ for all $v_k$ via testing extreme rays $\tilde{v}_k$ of polyhedral cones.
6:         Store extreme rays $\tilde{v}_k$ s.t. $g_k(z, \tilde{v}_k)^T \tilde{v}_k = 0$ for second-order test.
7:     **else**
8:         Test if $\partial_{[W_1\ b_1]_{k,\cdot}}\mathfrak{R} = \{\mathbf{0}_{d_x+1}^T\}$.
9:     **end if**
10: **end for**
11: For all $\eta$'s s.t. $g(z, \eta)^T \eta = 0$, test if $\eta^T H(z, \eta)\eta \geq 0$.
12: **if** $\exists \eta \neq \mathbf{0}$ s.t. $g(z, \eta)^T \eta = 0$ and $\eta^T H(z, \eta)\eta = 0$ **then**
13:     **return** SOSP.
14: **else**
15:     **return** Local Minimum.
16: **end if**

---

**Lemma 2.** *For small enough perturbation $\eta$,*
$$\mathfrak{R}(z + \eta) = \mathfrak{R}(z) + g(z, \eta)^T \eta + \tfrac{1}{2}\eta^T H(z, \eta)\eta + o(\|\eta\|^2),$$
*where $g(z, \eta)$ and $H(z, \eta)$ satisfy*

$$g(z, \eta)^T \eta = \sum_i \nabla \ell_i^T dY_1(x_i) = \left\langle \sum_i \nabla \ell_i O(x_i)^T, \Delta_2 \right\rangle + \left\langle \sum_i \nabla \ell_i, \delta_2 \right\rangle + \sum_{k=1}^{d_h} g_k(z, v_k)^T v_k,$$

$$\eta^T H(z, \eta)\eta = \sum_i \nabla \ell_i^T dY_2(x_i) + \tfrac{1}{2}\sum_i \|dY_1(x_i)\|_{\nabla^2 \ell_i}^2,$$

*and $g_k(z, v_k)^T := [W_2]_{\cdot,k}^T \left(C_k + \sum_{i \in B_k} h'(\bar{x}_i^T v_k)\nabla \ell_i \bar{x}_i^T\right)$. Also, $g(z, \eta)$ and $H(z, \eta)$ are piecewise constant functions of $\eta$, which are constant inside each polyhedral cone in space of $\eta$.*

Rough pseudocode of SOSP-CHECK is presented in Algorithm 1. As described in Section 2.1, the algorithm consists of three steps: (a) testing first-order stationarity (b) testing $g(z, \eta)^T \eta \geq 0$ for all $\eta$, and (c) testing $\eta^T H(z, \eta)\eta \geq 0$ for $\{\eta \mid g(z, \eta)^T \eta = 0\}$. If the input point satisfies the second-order sufficient conditions for local minimality, the algorithm decides it is a local minimum. If the point only satisfies second-order necessary conditions, it returns SOSP. If a strict descent direction $\eta$ is found, the algorithm terminates immediately and returns $\eta$. A brief description will follow, but the full algorithm (Algorithm 2) and a full proof of correctness are deferred to Appendix A.

### 3.1 TESTING FIRST-ORDER STATIONARITY (LINES 1, 4, AND 8)

Line 1 of Algorithm 1 corresponds to testing if $\partial_{W_2}\mathfrak{R}$ and $\partial_{b_2}\mathfrak{R}$ are singletons with zero. If not, the opposite direction is a descent direction. More details are in Appendix A.1.1.

Test for $W_1$ and $b_1$ is more difficult because $g(z, \eta)$ depends on $\Delta_1$ and $\delta_1$ when there are boundary data points. For each $k \in [d_h]$, Line 4 (if $M_k > 0$), and Line 8 (if $M_k = 0$) test if $\mathbf{0}_{d_x+1}^T$ is in $\partial_{[W_1\ b_1]_{k,\cdot}}\mathfrak{R}$. Note from Definition 2.1 and Lemma 2 that $\partial_{[W_1\ b_1]_{k,\cdot}}\mathfrak{R}$ is the convex hull of all possible values of $g_k(z, v_k)^T$. If $M_k > 0$, $\mathbf{0} \in \partial_{[W_1\ b_1]_{k,\cdot}}\mathfrak{R}$ can be tested by solving a convex QP:

$$\begin{array}{ll} \text{minimize}_{\{s_i\}_{i \in B_k}} & \|[W_2]_{\cdot,k}^T(C_k + \sum_{i \in B_k} s_i \nabla \ell_i \bar{x}_i^T)\|_2^2 \\ \text{subject to} & \min\{s_-, s_+\} \leq s_i \leq \max\{s_-, s_+\}, \ \forall i \in B_k. \end{array} \tag{2}$$

If the solution $\{s_i^*\}_{i \in B_k}$ does not achieve zero objective value, then we can directly return a descent direction. For details please refer to FO-SUBDIFF-ZERO-TEST (Algorithm 3) and Appendix A.1.2.

### 3.2 TESTING $g(z, \eta)^T \eta \geq 0$ FOR ALL $\eta$ (LINES 5–6)

**Linear program formulation.** Lines 5–6 are about testing if $g_k(z, v_k)^T v_k \geq 0$ for all directions of $v_k$. If $\mathbf{0}_{d_x+1}^T \in \partial_{[W_1\ b_1]_{k,\cdot}}\mathfrak{R}$, with the solution $\{s_i^*\}$ from QP (2) we can write $g_k(z, v_k)^T$ as

$$g_k(z, v_k)^T = [W_2]_{\cdot,k}^T\left(C_k + \sum_{i \in B_k} h'(\bar{x}_i^T v_k)\nabla \ell_i \bar{x}_i^T\right) = [W_2]_{\cdot,k}^T\left(\sum_{i \in B_k}\left(h'(\bar{x}_i^T v_k) - s_i^*\right)\nabla \ell_i \bar{x}_i^T\right).$$

Every $i \in B_k$ bisects $\mathbb{R}^{d_x+1}$ into two halfspaces, $\bar{x}_i^T v_k \geq 0$ and $\bar{x}_i^T v_k \leq 0$, in each of which $h'(\bar{x}_i^T v_k)$ stays constant. Note that by Lemma 1, $\bar{x}_i$'s for $i \in B_k$ are linearly independent. So, given $M_k$ boundary data points, they divide the space $\mathbb{R}^{d_x+1}$ of $v_k$ into $2^{M_k}$ polyhedral cones.

Since $g_k(z, v_k)^T$ is constant in each polyhedral cones, we can let $\sigma_i \in \{-1, +1\}$ for all $i \in B_k$, and define an LP for each $\{\sigma_i\}_{i \in B_k} \in \{-1, +1\}^{M_k}$:

$$\begin{aligned}
\underset{v_k}{\text{minimize}} \quad & [W_2]_{:,k}^T \left( \sum_{i \in B_k} (s_{\sigma_i} - s_i^*) \nabla \ell_i \bar{x}_i^T \right) v_k \\
\text{subject to} \quad & v_k \in \mathcal{V}_k, \quad \sigma_i \bar{x}_i^T v_k \geq 0, \ \forall i \in B_k.
\end{aligned} \quad (3)$$

Solving these LPs and checking if the minimum value is 0 suffices to prove $g_k(z, v_k)^T v_k \geq 0$ for all small enough perturbations. The constraint $v_k \in \mathcal{V}_k$ is there because any $v_k \notin \mathcal{V}_k$ is also orthogonal to $g_k(z, v_k)$. It is equivalent to $d_x + 1 - M_k$ linearly independent equality constraints. So, the feasible set of LP (3) has $d_x + 1$ linearly independent constraints, which implies that the feasible set is a pointed polyhedral cone with vertex at origin. Since any point in a pointed polyhedral cone is a conical combination (linear combination with nonnegative coefficients) of *extreme rays* of the cone, checking nonnegativity of the objective function for all extreme rays suffices. We emphasize that we *do not* solve the LPs (3) in our algorithm; we just check the extreme rays.

**Computational efficiency.** Extreme rays of a pointed polyhedral cone in $\mathbb{R}^{d_x+1}$ are computed from $d_x$ linearly independent active constraints. For each $i \in B_k$, the extreme ray $\hat{v}_{i,k} \in \mathcal{V}_k \cap \text{span}\{\bar{x}_j \mid j \in B_k \setminus \{i\}\}^{\perp}$ must be tested whether $g_k(z, \hat{v}_{i,k})^T \hat{v}_{i,k} \geq 0$, in both directions. Note that there are $2M_k$ extreme rays, and one extreme ray $\hat{v}_{i,k}$ is shared by $2^{M_k-1}$ polyhedral cones. Moreover, $\bar{x}_j^T \hat{v}_{i,k} = 0$ for $j \in B_k \setminus \{i\}$, which indicates that

$$g_k(z, \hat{v}_{i,k})^T \hat{v}_{i,k} = (s_{\sigma_{i,k}} - s_i^*)[W_2]_{:,k}^T \nabla \ell_i \bar{x}_i^T \hat{v}_{i,k}, \text{ where } \sigma_{i,k} = \text{sign}(\bar{x}_i^T \hat{v}_{i,k}),$$

regardless of $\{\sigma_j\}_{j \in B_k \setminus \{i\}}$. Testing an extreme ray can be done with a *single* inequality test instead of $2^{M_k-1}$ separate tests for all cones! Thus, this extreme ray approach instead of solving individual LPs greatly reduces computation, from $O(2^{M_k})$ to $O(M_k)$.

**Testing extreme rays.** For the details of testing all possible extreme rays, please refer to FO-INCREASING-TEST (Algorithm 4) and Appendix A.2. FO-INCREASING-TEST computes all possible extreme rays $\tilde{v}_k$ and tests if they satisfy $g_k(z, \tilde{v}_k)^T \tilde{v}_k \geq 0$. If the inequality is not satisfied by an extreme ray $\tilde{v}_k$, then this is a descent direction, so we return $\tilde{v}_k$. If the inequality holds with equality, it means this is a *flat extreme ray*, and it needs to be checked in second-order test, so we save this extreme ray for future use.

How many flat extreme rays ($g_k(z, \tilde{v}_k)^T \tilde{v}_k = 0$) are there? Presence of flat extreme rays introduce inequality constraints in the QP that we solve in the second-order test. It is ideal not to have them, because in this case there are only equality constraints, so the QP is easier to solve. Lemma A.1 in Appendix A.2 shows the conditions for having flat extreme rays; in short, there is a flat extreme ray if $[W_2]_{:,k}^T \nabla \ell_i = 0$ or $s_i^* = s_+$ or $s_-$. For more details, please refer to Appendix A.2.

## 3.3 TESTING $\eta^T H(z, \eta) \eta \geq 0$ FOR $\{\eta \mid g(z, \eta)^T \eta = 0\}$ (LINES 11–16)

The second-order test checks $\eta^T H(z, \eta) \eta \geq 0$ for "flat" $\eta$'s satisfying $g(z, \eta)^T \eta = 0$. This is done with help of the function SO-TEST (Algorithm 5). Given its input $\{\sigma_{i,k}\}_{k \in [d_h], i \in B_k}$, it defines fixed "Jacobian" matrices $J_i$ for all data points and equality/inequality constraints for boundary data points, and solves the QP of the following form:

$$\begin{aligned}
\text{minimize}_{\eta} \quad & \sum_i \nabla \ell_i^T \Delta_2 J_i (\Delta_1 x_i + \delta_1) + \tfrac{1}{2} \sum_i \| \Delta_2 O(x_i) + \delta_2 + W_2 J_i (\Delta_1 x_i + \delta_1) \|_{\nabla^2 \ell_i}^2, \\
\text{subject to} \quad & [W_2]_{:,k}^T u_k = [W_1 \ b_1]_{k,.} v_k, \quad \forall k \in [d_h], \\
& \bar{x}_i^T v_k = 0, \quad \forall k \in [d_h], \forall i \in B_k \text{ s.t. } \sigma_{i,k} = 0, \\
& \sigma_{i,k} \bar{x}_i^T v_k \geq 0, \quad \forall k \in [d_h], \forall i \in B_k \text{ s.t. } \sigma_{i,k} \in \{-1, +1\}.
\end{aligned} \quad (4)$$

**Constraints and number of QPs.** There are $d_h$ equality constraints of the form $[W_2]_{:,k}^T u_k = [[W_1]_{k,.} \ [b_1]_k] v_k$. These equality constraints are due to the nonnegative homogeneous property of activation $h$; i.e., scaling $[W_1]_{k,.}$ and $[b_1]_k$ by $\alpha > 0$ and scaling $[W_2]_{:,k}$ by $1/\alpha$ yields exactly

the same network. So, these equality constraints force $\eta$ to be orthogonal to the loss-invariant directions. This observation is stated more formally in Lemma A.2, which as a corollary shows that any differentiable FOSP of $\mathfrak{R}$ always has rank-deficient Hessian. The other constraints make sure that the union of feasible sets of QPs is exactly $\{\eta \mid g(z, \eta)^T \eta = 0\}$ (please see Lemma A.3 in Appendix A.3 for details). It is also easy to check that these constraints are all linearly independent.

If there is no flat extreme ray, the algorithm solves just one QP with $d_h + M$ equality constraints. If there are flat extreme rays, the algorithm solves one QP with $d_h + M$ equality constraints, and $2^K$ more QPs with $d_h + M - L$ equality constraints and $L$ inequality constraints, where

$$K := \sum_{k=1}^{d_h} \left| \{i \in B_k \mid [W_2]_{\cdot,k}^T \nabla \ell_i = 0\} \right|, \quad L := \sum_{k=1}^{d_h} \left| \{i \in B_k \mid \hat{v}_{i,k} \text{ or } -\hat{v}_{i,k} \text{ is a flat ext. ray}\} \right|. \quad (5)$$

Recall from Section 3.2 that $i \in B_k$ has a flat extreme ray if $[W_2]_{\cdot,k}^T \nabla \ell_i = 0$ or $s_i^* = s_+$ or $s_-$; thus, $K \leq L \leq M$. Please refer to Appendix A.3 for more details.

**Efficiency of solving the QPs** (4). Despite NP-hardness of general QPs, our specific form of QPs (4) can be solved quite efficiently, avoiding exponential complexity in $p$. After solving QP (4), there are three (disjoint) termination conditions:

(T1) $\eta^T Q \eta > 0$ whenever $\eta \in \mathcal{S}, \eta \neq \mathbf{0}$, or
(T2) $\eta^T Q \eta \geq 0$ whenever $\eta \in \mathcal{S}$, but $\exists \eta \neq \mathbf{0}, \eta \in \mathcal{S}$ such that $\eta^T Q \eta = 0$, or
(T3) $\exists \eta$ such that $\eta \in \mathcal{S}$ and $\eta^T Q \eta < 0$,

where $\mathcal{S}$ is the feasible set of QP. With the following two lemmas, we show that the termination conditions can be efficiently tested for ECQPs and ICQPs. First, the ECQPs can be iteratively solved with projected gradient descent, as stated in the next lemma.

**Lemma 3.** *Consider the QP, where $Q \in \mathbb{R}^{p \times p}$ is symmetric and $A \in \mathbb{R}^{q \times p}$ has full row rank:*

$$\text{minimize}_\eta \quad \tfrac{1}{2} \eta^T Q \eta \quad \text{subject to} \quad A\eta = \mathbf{0}_q$$

*Then, projected gradient descent (PGD) updates*

$$\eta^{(t+1)} = (I - A^T(AA^T)^{-1}A)(I - \alpha Q)\eta^{(t)}$$

*with learning rate $\alpha < 1/\lambda_{\max}(Q)$ converges to a solution or diverges to infinity exponentially fast. Moreover, with random initialization, PGD correctly checks conditions (T1)–(T3) with probability* 1.

The proof is an extension of unconstrained case (Lee et al., 2016), and is deferred to Appendix B.3. Note that it takes $O(p^2 q)$ time to compute $(I - A^T(AA^T)^{-1}A)(I - \alpha Q)$ in the beginning, and each update takes $O(p^2)$ time. It is also surprising that the convergence rate does not depend on $q$.

In the presence of flat extreme rays, we have to solve QPs involving $L$ inequality constraints. We prove that our ICQP can be solved in $O(p^3 + L^3 2^L)$ time, which implies that as long as the number of flat extreme rays is small, the problem can still be solved in polynomial time in $p$.

**Lemma 4.** *Consider the QP, where $Q \in \mathbb{R}^{p \times p}$ is symmetric, $A \in \mathbb{R}^{q \times p}$ and $B \in \mathbb{R}^{r \times p}$ have full row rank, and $\begin{bmatrix} A^T & B^T \end{bmatrix}$ has rank $q + r$:*

$$\text{minimize}_\eta \quad \eta^T Q \eta \quad \text{subject to} \quad A\eta = \mathbf{0}_q, \ B\eta \geq \mathbf{0}_r.$$

*Then, there exists a method that checks whether (T1)–(T3) in $O(p^3 + r^3 2^r)$ time.*

In short, we transform $\eta$ to define an equivalent problem, and use classical results in copositive matrices (Martin & Jacobson, 1981; Seeger, 1999; Hiriart-Urruty & Seeger, 2010); the problem can be solved by computing the eigensystem of a $(p-q-r) \times (p-q-r)$ matrix, and testing copositivity of an $r \times r$ matrix. The proof is presented in Appendix B.4.

**Concluding the test.** During all calls to SO-TEST, whenever any QP terminated with (T3), then SOSP-CHECK immediately returns the direction and terminates. After solving all QPs, if any of SO-TEST calls finished with (T2), then we conclude SOSP-CHECK with "SOSP." If all QPs terminated with (T1), then we can return "Local Minimum."

Table 1: Summary of experimental results

| $(d_x, d_h, m)$ | # Runs | Sum $M$ (Avg.) | Sum $L$ (Avg.) | Sum $K$ (Avg.) | $\mathbb{P}\{L > 0\}$ |
|---|---|---|---|---|---|
| $(10, 1, 1000)$ | 40 | 290 (7.25) | 0 (0) | 0 (0) | 0 |
| $(10, 1, 10000)$ | 40 | 371 (9.275) | 1 (0.025) | 0 (0) | 0.025 |
| $(100, 1, 1000)$ | 40 | 1,452 (36.3) | 0 (0) | 0 (0) | 0 |
| $(100, 1, 10000)$ | 40 | 2,976 (74.4) | 2 (0.05) | 0 (0) | 0.05 |
| $(100, 10, 10000)$ | 40 | 24,805 (620.125) | 4 (0.1) | 0 (0) | 0.1 |
| $(1000, 1, 10000)$ | 40 | 14,194 (354.85) | 0 (0) | 0 (0) | 0 |
| $(1000, 10, 10000)$ | 40 | 42,334 (1,058.35) | 37 (0.925) | 1 (0.025) | 0.625 |

## 4 Experiments

For experiments, we used artificial datasets sampled iid from standard normal distribution, and trained 1-hidden-layer ReLU networks with squared error loss. In practice, it is impossible to get to the exact nondifferentiable point, because they lie in a set of measure zero. To get close to those points, we ran Adam (Kingma & Ba, 2014) using full-batch (exact) gradient for 200,000 iterations and decaying step size (start with $10^{-3}$, $0.2\times$ decay every 20,000 iterations). We observed that decaying step size had the effect of "descending deeper into the valley."

After running Adam, for each $k \in [d_h]$, we counted the number of *approximate boundary data points* satisfying $|[W_1 x_i + b_1]_k| < 10^{-5}$, which gives an estimate of $M_k$. Moreover, for these points, we solved the QP (2) using L-BFGS-B (Byrd et al., 1995), to check if the terminated points are indeed (approximate) FOSPs. We could see that the optimal values of (2) are close to zero ($\leq 10^{-6}$ typically, $\leq 10^{-3}$ for largest problems). After solving (2), we counted the number of $s_i^*$'s that ended up with 0 or 1. The number of such $s_i^*$'s is an estimate of $L - K$. We also counted the number of approximate boundary data points satisfying $|[W_2]_{\cdot,k}^T \nabla \ell_i| < 10^{-4}$, for an estimate of $K$.

We ran the above-mentioned experiments for different settings of $(d_x, d_h, m)$, 40 times each. We fixed $d_y = 1$ for simplicity. For large $d_h$, the optimizer converged to near-zero minima, making $\nabla \ell_i$ uniformly small, so it was difficult to obtain accurate estimates of $K$ and $L$. Thus, we had to perform experiments in settings where the optimizer converged to minima that are far from zero.

Table 1 summarizes the results. Through 280 runs, we observed that there are surprisingly many boundary data points ($M$) in general, but usually there are zero or very few (maximum was 3) flat extreme rays ($L$). This observation suggests two important messages: (1) many local minima are on nondifferentiable points, which is the reason why our analysis is meaningful; (2) luckily, $L$ is usually very small, so we only need to solve ECQPs ($L = 0$) or ICQPs with very small number of inequality constraints, which are solved efficiently (Lemmas 3 and 4). We can observe that $M$, $L$, and $K$ indeed increase as model dimensions and training set get larger, but the rate of increase is not as fast as $d_x$, $d_h$, and $m$.

## 5 Discussion and future work

We provided a theoretical algorithm that tests second-order stationarity and escapes saddle points, for any points (including nondifferentiable ones) of empirical risk of shallow ReLU-like networks. Despite difficulty raised by boundary data points dividing the parameter space into $2^M$ regions, we reduced the computation to $d_h$ convex QPs, $O(M)$ equality/inequality tests, and one (or a few more) nonconvex QP. In benign cases, the last QP is equality constrained, which can be efficiently solved with projected gradient descent. In worse cases, the QP has a few (say $L$) inequality constraints, but it can be solved efficiently when $L$ is small. We also provided empirical evidences that $L$ is usually either zero or very small, suggesting that the test can be done efficiently in most cases. A limitation of this work is that in practice, *exact* nondifferentiable points are impossible to reach, so the algorithm must be extended to apply the nonsmooth analysis for points that are "close" to nondifferentiable ones. Also, current algorithm only tests for *exact* SOSP, while it is desirable to check approximate second-order stationarity. These extensions must be done in order to implement a robust numerial version of the algorithm, but they require significant amount of additional work; thus, we leave practical/robust implementation as future work. Also, extending the test to deeper neural networks is an interesting future direction.

ACKNOWLEDGMENTS

This work was supported by the DARPA Lagrange Program. Suvrit Sra also acknowledges support from an Amazon Research Award.

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

---

**Algorithm 2** SOSP-CHECK

---

Input: A tuple $(W_j, b_j)_{j=1}^2$ of $\mathfrak{R}(\cdot)$.
1: **if** $\sum_{i=1}^m \nabla \ell_i \left[ O(x_i)^T \quad 1 \right] \neq \mathbf{0}_{d_y \times (d_h+1)}$ **then**
2:     **return** $[\Delta_2 \quad \delta_2] \leftarrow - \sum_{i=1}^m \nabla \ell_i \left[ O(x_i)^T \quad 1 \right]$, $\Delta_1 \leftarrow \mathbf{0}_{d_h \times d_x}$, $\delta_1 \leftarrow \mathbf{0}_{d_h}$.
3: **end if**
4: **for** $k \in [d_h]$ **do**
5:     **if** $M_k > 0$ **then**
6:         $\{s_i^*\}_{i \in B_k} \leftarrow$ FO-SUBDIFF-ZERO-TEST$(k)$
7:         $\tilde{v}_k^T \leftarrow [W_2]_{\cdot,k}^T (C_k + \sum_{i \in B_k} s_i^* \nabla \ell_i \bar{x}_i^T)$.
8:         **if** $\tilde{v}_k \neq \mathbf{0}_{d_x+1}$ **then**
9:             **return** $v_k \leftarrow -\tilde{v}_k$, $\forall k' \in [d_h] \setminus \{k\}, v_{k'} \leftarrow \mathbf{0}_{d_x+1}$, $\Delta_2 \leftarrow \mathbf{0}_{d_y \times d_h}$, $\delta_2 \leftarrow \mathbf{0}_{d_y}$.
10:         **end if**
11:         $(\texttt{decr}, \tilde{v}_k, \{S_{i,k}\}_{i \in B_k}) \leftarrow$ FO-INCREASING-TEST$(k, \{s_i^*\}_{i \in B_k})$.
12:         **if** $\texttt{decr} = \texttt{True}$ **then**
13:             **return** $v_k \leftarrow \tilde{v}_k$, $\forall k' \in [d_h] \setminus \{k\}, v_{k'} \leftarrow \mathbf{0}_{d_x+1}$, $\Delta_2 \leftarrow \mathbf{0}_{d_y \times d_h}$, $\delta_2 \leftarrow \mathbf{0}_{d_y}$.
14:         **end if**
15:     **else if** $[W_2]_{\cdot,k}^T C_k \neq \mathbf{0}_{d_x+1}^T$ **then**
16:         **return** $v_k \leftarrow -C_k^T [W_2]_{\cdot,k}$, $\forall k' \in [d_h] \setminus \{k\}, v_{k'} \leftarrow \mathbf{0}_{d_x+1}$, $\Delta_2 \leftarrow \mathbf{0}_{d_y \times d_h}$, $\delta_2 \leftarrow \mathbf{0}_{d_y}$.
17:     **end if**
18: **end for**
19: $(\texttt{decr}, \texttt{sosp}, (\Delta_j, \delta_j)_{j=1}^2) \leftarrow$ SO-TEST$(\{0\}_{k \in [d_h], i \in B_k})$.
20: **if** $\texttt{decr} = \texttt{True}$ **then return** $(\Delta_j, \delta_j)_{j=1}^2$.
21: **end if**
22: **if** $M \neq 0$ and $\{S_{i,k}\}_{k \in [d_h], i \in B_k} \neq \{\{0\}\}_{k \in [d_h], i \in B_k}$ **then**
23:     **for** each element $\{\sigma_{i,k}\}_{k \in [d_h], i \in B_k} \in \prod_{k \in [d_h]} \prod_{i \in B_k} S_{i,k}$ **do**
24:         $(\texttt{decr}, \texttt{sospTemp}, (\Delta_j, \delta_j)_{j=1}^2) \leftarrow$ SO-TEST$(\{\sigma_{i,k}\}_{k \in [d_h], i \in B_k})$.
25:         **if** $\texttt{decr} = \texttt{True}$ **then return** $(\Delta_j, \delta_j)_{j=1}^2$.
26:         **end if**
27:         $\texttt{sosp} \leftarrow \texttt{sosp} \vee \texttt{sospTemp}$
28:     **end for**
29: **end if**
30: **if** $\texttt{sosp} = \texttt{True}$ **then return** SOSP.
31: **else return** Local Minimum.
32: **end if**

---

**Algorithm 3** FO-SUBDIFF-ZERO-TEST

---

Input: $k \in [d_h]$
1: Solve the following optimization problem and get optimal solution $\{s_i^*\}_{i \in B_k}$:

$$\begin{array}{ll} \text{minimize}_{\{s_i\}_{i \in B_k}} & \|[W_2]_{\cdot,k}^T (C_k + \sum_{i \in B_k} s_i \nabla \ell_i \bar{x}_i^T)\|_2^2 \\ \text{subject to} & \min\{s_-, s_+\} \leq s_i \leq \max\{s_-, s_+\}, \ \forall i \in B_k, \end{array} \quad (2)$$

2: **return** $\{s_i^*\}_{i \in B_k}$.

---

## A    FULL ALGORITHMS AND PROOF OF CORRECTNESS

In this section, we present the detailed operation of SOSP-CHECK (Algorithm 2), and its helper functions FO-SUBDIFF-ZERO-TEST, FO-INCREASING-TEST, and SO-TEST (Algorithm 3–5).

In the subsequent subsections, we provide a more detailed proof of the correctness of Algorithm 2. Recall that, by Lemmas 1 and 2, $M_k := |B_k| \leq d_x$ and vectors $\{\bar{x}_i\}_{i \in B_k}$ are linearly independent. Also, we can expand $\mathfrak{R}(z + \eta)$ so that

$$\mathfrak{R}(z + \eta) = \mathfrak{R}(z) + g(z, \eta)^T \eta + \tfrac{1}{2} \eta^T H(z, \eta) \eta + o(\|\eta\|^2),$$

---

**Algorithm 4** FO-INCREASING-TEST

Input: $k \in [d_h]$, $\{s_i^*\}_{i \in B_k}$

1: **for all** $i \in B_k$ **do**
2:      Define $S_{i,k} \leftarrow \emptyset$.
3:      Get a vector $\hat{v}_{i,k} \in \mathcal{V}_k \cap \mathrm{span}\{\bar{x}_j \mid j \in B_k \setminus \{i\}\}^\perp$.
4:      **for** $\tilde{v}_k \in \{\hat{v}_{i,k}, -\hat{v}_{i,k}\}$ **do**
5:          Define $\sigma_{i,k} \leftarrow \mathrm{sign}(\bar{x}_i^T \tilde{v}_k)$.
6:          **if** $(s_{\sigma_{i,k}} - s_i^*)[W_2]_{\cdot,k}^T \nabla \ell_i \bar{x}_i^T \tilde{v}_k < 0$ **then**
7:              **return** $(\texttt{True}, \tilde{v}_k, \{\emptyset\}_{i \in B_k})$
8:          **else if** $(s_{\sigma_{i,k}} - s_i^*)[W_2]_{\cdot,k}^T \nabla \ell_i \bar{x}_i^T \tilde{v}_k = 0$ **then**
9:              $S_{i,k} \leftarrow S_{i,k} \cup \{\sigma_{i,k}\}$.
10:          **end if**
11:      **end for**
12:      If $S_{i,k} = \emptyset$, $S_{i,k} \leftarrow \{0\}$.
13: **end for**
14: **return** $(\texttt{False}, \mathbf{0}_{d_x+1}, \{S_{i,k}\}_{i \in B_k})$.

---

**Algorithm 5** SO-TEST

Input: $\{\sigma_{i,k}\}_{k \in [d_h], i \in B_k}$

1: For all $i \in [m]$, define diagonal matrices $J_i \in \mathbb{R}^{d_h \times d_h}$ such that for $k \in [d_h]$,

$$[J_i]_{k,k} \leftarrow \begin{cases} h'([N(x_i)]_k) & \text{if } i \in [m] \setminus B_k \\ s_{\sigma_{i,k}} & \text{if } i \in B_k \text{ and } \sigma_{i,k} \in \{-1, +1\} \\ 0 & \text{if } i \in B_k \text{ and } \sigma_{i,k} = 0. \end{cases}$$

2: Solve the following QP. If there is no solution, get a descent direction $(\Delta_j^*, \delta_j^*)_{j=1}^2$.

$$\begin{aligned} \underset{\eta}{\text{minimize}} \quad & \sum_i \nabla \ell_i^T \Delta_2 J_i (\Delta_1 x_i + \delta_1) + \tfrac{1}{2} \sum_i \|\Delta_2 O(x_i) + \delta_2 + W_2 J_i (\Delta_1 x_i + \delta_1)\|_{\nabla^2 \ell_i}^2, \\ \text{subject to} \quad & [W_2]_{\cdot,k}^T u_k = [W_1 \ \ b_1]_{k,\cdot} v_k, && \forall k \in [d_h], \\ & \bar{x}_i^T v_k = 0, && \forall k \in [d_h], \forall i \in B_k \text{ s.t. } \sigma_{i,k} = 0, \\ & \sigma_{i,k} \bar{x}_i^T v_k \ge 0, && \forall k \in [d_h], \forall i \in B_k \text{ s.t. } \sigma_{i,k} \in \{-1, +1\}. \end{aligned} \quad (4)$$

3: **if** There is no solution **then return** $(\texttt{True}, \texttt{False}, (\Delta_j^*, \delta_j^*)_{j=1}^2)$.
4: **else if** QP has nonzero minimizers **then return** $(\texttt{False}, \texttt{True}, \mathbf{0})$
5: **else return** $(\texttt{False}, \texttt{False}, \mathbf{0})$
6: **end if**

---

where $g(z, \eta)$ and $H(z, \eta)$ satisfy

$$g(z, \eta)^T \eta = \sum_i \nabla \ell_i^T dY_1(x_i) = \left\langle \sum_i \nabla \ell_i O(x_i)^T, \Delta_2 \right\rangle + \left\langle \sum_i \nabla \ell_i, \delta_2 \right\rangle + \sum_{k=1}^{d_h} g_k(z, v_k)^T v_k,$$

$$\eta^T H(z, \eta) \eta = \sum_i \nabla \ell_i^T dY_2(x_i) + \tfrac{1}{2} \sum_i \|dY_1(x_i)\|_{\nabla^2 \ell_i}^2,$$

and $g_k(z, v_k)^T := [W_2]_{\cdot,k}^T \left( C_k + \sum_{i \in B_k} h'(\bar{x}_i^T v_k) \nabla \ell_i \bar{x}_i^T \right)$.

## A.1 TESTING FIRST-ORDER STATIONARITY (LINES 1–3, 6–10 AND 15–17)

### A.1.1 TEST OF FIRST-ORDER STATIONARITY FOR $W_2$ AND $b_2$ (LINES 1–3)

Lines 1–3 of Algorithm 2 correspond to testing if $\partial_{W_2} \mathfrak{R} = \{\mathbf{0}_{d_y \times d_h}\}$ and $\partial_{b_2} \mathfrak{R} = \{\mathbf{0}_{d_y}\}$. If they are not all zero, the opposite direction is a descent direction, as Line 2 returns. To see why, suppose $\sum_{i=1}^m \nabla \ell_i [O(x_i)^T \ \ 1] \ne \mathbf{0}_{d_y \times (d_h+1)}$. Then choose perturbations

$$[\Delta_2 \ \ \delta_2] = -\sum_{i=1}^m \nabla \ell_i [O(x_i)^T \ \ 1], \ \ \Delta_1 = \mathbf{0}_{d_h \times d_x}, \ \ \delta_1 = \mathbf{0}_{d_h}.$$

If we apply perturbation $(\gamma \Delta_j, \gamma \delta_j)_{j=1}^2$ where $\gamma > 0$, we can immediately check that $dY_1(x_i) = \Delta_2 O(x_i) + \delta_2$ and $dY_2(x_i) = \mathbf{0}$. So,

$$g(z, \eta)^T \eta = \sum_{i=1}^m \nabla \ell_i^T \left( \Delta_2 O(x_i) + \delta_2 \right) = \left\langle \sum_{i=1}^m \nabla \ell_i \left[ O(x_i)^T \quad 1 \right], \left[ \Delta_2 \quad \delta_2 \right] \right\rangle = -O(\gamma),$$

$$\eta^T H(z, \eta) \eta = \frac{1}{2} \sum_i dY_1(x_i)^T \nabla^2 \ell_i dY_1(x_i) = O(\gamma^2) \geq 0.$$

and also that $\sum_{i=1}^m \| dY(x_i) \|_2^2 = O(\gamma^2)$. Then, by scaling $\gamma$ sufficiently small we can achieve $\mathfrak{R}(z + \eta) < \mathfrak{R}(z)$, which disproves that $(W_j, b_j)_{j=1}^2$ is a local minimum.

### A.1.2 Test of first-order stationarity for $W_1$ and $b_1$ (lines 6–10 and 15–17)

Test for $W_1$ and $b_1$ is more difficult because $g(z, \eta)$ depends on $\Delta_1$ and $\delta_1$ when there are boundary data points. Recall that $v_k^T$ ($k \in [d_h]$) is the $k$-th row of $[\Delta_1 \quad \delta_1]$. Then note from Lemma 2 that

$$\sum_{i=1}^m \nabla \ell_i^T \left( W_2 J(x_i)(\Delta_1 x_i + \delta_1) \right) = \sum_{k=1}^{d_h} g_k(z, v_k)^T v_k,$$

where $g_k(z, v_k)^T := [W_2]_{\cdot,k}^T \left( C_k + \sum_{i \in B_k} h'(\bar{x}_i^T v_k) \nabla \ell_i \bar{x}_i^T \right)$. Thus we can separate $k$'s and treat them individually.

**Test for zero gradient.** Recall the definition $M_k := |B_k|$. If $M_k = 0$, there is no boundary data point for $k$-th hidden node, so the Clarke subdifferential with respect to $[W_1 \ b_1]_{k,\cdot}$ is $\{C_k^T [W_2]_{\cdot,k}\}$. Lines 15–17 handle this case; if the singleton element in the subdifferential is not zero, its opposite direction is a descent direction, so return that direction, as in Line 16.

**Test for zero in subdifferential.** For the case $M_k > 0$, we saw that for boundary data points $i \in B_k$, $h'([\Delta_1 x_i + \delta_1]_k) = h'(\bar{x}_i^T v_k) \in \{s_-, s_+\}$ depends on $v_k$. Lines 6–10 test if $\mathbf{0}_{d_x+1}^T$ is in the Clarke subdifferential of $\mathfrak{R}$ with respect to $[W_1]_{k,\cdot}$ and $[b_1]_k$. Since the subdifferential is used many times, we give it a specific name $\mathcal{D}_k := \partial_{[W_1 \ b_1]_{k,\cdot}} \mathfrak{R}$. By observing that $\mathcal{D}_k$ is the convex hull of all possible values of $g_k(z, v_k)^T$,

$$\mathcal{D}_k := \left\{ [W_2]_{\cdot,k}^T \left( C_k + \sum_{i \in B_k} s_i \nabla \ell_i \bar{x}_i^T \right) \mid \min\{s_-, s_+\} \leq s_i \leq \max\{s_-, s_+\}, \ \forall i \in B_k \right\}.$$

Testing $\mathbf{0}_{d_x+1}^T \in \mathcal{D}_k$ is done by FO-SUBDIFF-ZERO-TEST in Algorithm 3. It solves a convex QP (2), and returns $\{s_i^*\}_{i \in B_k}$.

If $\mathbf{0}_{d_x+1}^T \in \mathcal{D}_k$, $\{s_i^*\}_{i \in B_k}$ will satisfy $\tilde{v}_k^T := [W_2]_{\cdot,k}^T (C_k + \sum_{i \in B_k} s_i^* \nabla \ell_i \bar{x}_i^T) = \mathbf{0}_{d_x+1}^T$. Suppose $\mathbf{0}_{d_x+1}^T \notin \mathcal{D}_k$. Then, $\tilde{v}_k$ is the closest vector in $\mathcal{D}_k$ from the origin, so $\langle \tilde{v}_k, v \rangle > 0$ for all $v^T \in \mathcal{D}_k$. Choose perturbations

$$v_k = -\tilde{v}_k, \ \ v_{k'} = \mathbf{0}_{d_x+1} \text{ for all } k' \in [d_h] \setminus \{k\}, \ \ \Delta_2 = \mathbf{0}_{d_y \times d_h}, \ \ \delta_2 = \mathbf{0}_{d_y},$$

and apply perturbation $(\gamma \Delta_j, \gamma \delta_j)_{j=1}^2$ where $\gamma > 0$. With this perturbation, we can check that

$$g(z, \eta)^T \eta = \sum_{i=1}^m \nabla \ell_i^T dY_1(x_i) = -\gamma [W_2]_{\cdot,k}^T \left( C_k + \sum_{i \in B_k} h'(-\bar{x}_i^T \tilde{v}_k) \nabla \ell_i \bar{x}_i^T \right) \tilde{v}_k,$$

and since $h'(-\bar{x}_i^T \tilde{v}_k) \in \{s_-, s_+\}$ for $i \in B_k$, we have

$$[W_2]_{\cdot,k}^T \left( C_k + \sum_{i \in B_k} h'(-\bar{x}_i^T \tilde{v}_k) \nabla \ell_i \bar{x}_i^T \right) \in \mathcal{D}_k,$$

and $\langle \tilde{v}_k, v \rangle > 0$ for all $v^T \in \mathcal{D}_k$ shows that $g(z, \eta)^T \eta$ is strictly negative with magnitude $O(\gamma)$. It is easy to see that $\eta^T H(z, \eta) \eta = O(\gamma^2)$, so by scaling $\gamma$ sufficiently small we can disprove local minimality of $(W_j, b_j)_{j=1}^2$.

## A.2 Testing $g(z, \eta)^T \eta \geq 0$ for all $\eta$ (lines 11–14)

**Linear program formulation.** Lines 11–14 are essentially about testing if $g_k(z, v_k)^T v_k \geq 0$ for all directions $v_k$. If $\mathbf{0}_{d_x+1}^T \in \mathcal{D}_k$, with the solution $\{s_i^*\}_{i \in B_k}$ from FO-Subdiff-Zero-Test we can write $g_k(z, v_k)^T$ as

$$g_k(z, v_k)^T = [W_2]_{\cdot,k}^T \left( C_k + \sum_{i \in B_k} h'(\bar{x}_i^T v_k) \nabla \ell_i \bar{x}_i^T \right) = [W_2]_{\cdot,k}^T \left( \sum_{i \in B_k} \left( h'(\bar{x}_i^T v_k) - s_i^* \right) \nabla \ell_i \bar{x}_i^T \right).$$

For any $i \in B_k$, $h'(\bar{x}_i^T v_k) \in \{s_-, s_+\}$ changes whenever the sign of $\bar{x}_i^T v_k$ changes. Every $i \in B_k$ bisects $\mathbb{R}^{d_x+1}$ into two halfspaces, $\bar{x}_i^T v_k \geq 0$ and $\bar{x}_i^T v_k \leq 0$, in each of which $h'(\bar{x}_i^T v_k)$ stays constant. Note that by Lemma 1, $\bar{x}_i$'s for $i \in B_k$ are linearly independent. So, given $M_k$ linearly independent $\bar{x}_i$'s, they divide the space $\mathbb{R}^{d_x+1}$ of $v_k$ into $2^{M_k}$ polyhedral cones.

Since $g_k(z, v_k)^T$ is constant in each polyhedral cone, we can let $\sigma_i \in \{-1, +1\}$ for all $i \in B_k$, and define an LP for each $\{\sigma_i\}_{i \in B_k} \in \{-1, +1\}^{M_k}$:

$$
\begin{aligned}
\underset{v_k}{\text{minimize}} \quad & [W_2]_{\cdot,k}^T \left( \sum_{i \in B_k} (s_{\sigma_i} - s_i^*) \nabla \ell_i \bar{x}_i^T \right) v_k \\
\text{subject to} \quad & v_k \in \mathcal{V}_k, \quad \sigma_i \bar{x}_i^T v_k \geq 0, \ \forall i \in B_k.
\end{aligned}
\tag{3}
$$

Solving these LPs and checking if the minimum value is 0 suffices to prove $g_k(z, v_k)^T v_k \geq 0$ for all small enough perturbations. Recall that $\mathcal{V}_k := \text{span}\{\bar{x}_i \mid i \in B_k\}$ and $\dim(\mathcal{V}_k) = M_k$. Note that any component of $v_k$ that is orthogonal to $\mathcal{V}_k$ is also orthogonal to $g_k(z, v_k)$, so it does not affect the objective function of any LP (3). Thus, the constraint $v_k \in \mathcal{V}_k$ is added to the LP (3), which is equivalent to adding $d_x + 1 - M_k$ linearly independent equality constraints. The feasible set of LP (3) has $d_x + 1$ linearly independent equality/inequality constraints, which implies that the feasible set is a pointed polyhedral cone with vertex at origin. Since any point in a pointed polyhedral cone is a conical combination (linear combination with nonnegative coefficients) of *extreme rays* of the cone, checking nonnegativity of the objective function for all *extreme rays* suffices. We emphasize that we *do not* solve the LPs (3) in our algorithm; we just check the extreme rays.

**Computational efficiency.** Extreme rays of a pointed polyhedral cone in $\mathbb{R}^{d_x+1}$ are computed from $d_x$ linearly independent active constraints. Line 3 of Algorithm 4 is exactly computing such extreme rays: $\hat{v}_{i,k} \in \mathcal{V}_k \cap \text{span}\{\bar{x}_j \mid j \in B_k \setminus \{i\}\}^\perp$ for each $i \in B_k$, tested in both directions.

Note that there are $2M_k$ extreme rays, and one extreme ray $\hat{v}_{i,k}$ is shared by $2^{M_k-1}$ polyhedral cones. Moreover, $\bar{x}_j^T \hat{v}_{i,k} = 0$ for $j \in B_k \setminus \{i\}$, which indicates that

$$g_k(z, \hat{v}_{i,k})^T \hat{v}_{i,k} = (s_{\sigma_{i,k}} - s_i^*)[W_2]_{\cdot,k}^T \nabla \ell_i \bar{x}_i^T \hat{v}_{i,k}, \text{ where } \sigma_{i,k} = \text{sign}(\bar{x}_i^T \hat{v}_{i,k}),$$

regardless of $\{\sigma_j\}_{j \in B_k \setminus \{i\}}$. This observation is used in Lines 6 and 8 of Algorithm 4. Testing $g_k(z, \tilde{v}_k)^T \tilde{v}_k \geq 0$ for an extreme ray $\tilde{v}_k$ can be done with a *single* inequality test instead of $2^{M_k-1}$ separate tests for all cones! Thus, this extreme ray approach instead of solving individual LPs greatly reduces computation, from $O(2^{M_k})$ to $O(M_k)$.

**Algorithm operation in detail.** Testing all possible extreme rays is exactly what FO-Increasing-Test in Algorithm 4 is doing. Output of FO-Increasing-Test is a tuple of three items: a boolean, a $(d_x + 1)$-dimensional vector, and a tuple of $M_k$ sets. Whenever we have a descent direction, it returns True and the descent direction $\tilde{v}_k$. If there is no descent direction, it returns False and the sets $\{S_{i,k}\}_{i \in B_k}$.

For both direction of extreme rays $\tilde{v}_k = \hat{v}_{i,k}$ and $\tilde{v}_k = -\hat{v}_{i,k}$ (Line 4), we check if $g_k(z, \tilde{v}_k)^T \tilde{v}_k \geq 0$. Whenever it does not hold (Lines 6–7), $\tilde{v}_k$ is a descent direction, so FO-Increasing-Test returns it with True. Line 13 of Algorithm 2 uses that $\tilde{v}_k$ to return perturbations, so that scaling by small enough $\gamma > 0$ will give us a point with $\Re(z + \gamma \eta) < \Re(z)$. If equality holds (Lines 8–9), this means $\tilde{v}_k$ is a direction of perturbation satisfying $g(z, \eta)^T \eta = 0$, so this direction needs to be checked if $\eta^T H(z, \eta) \eta \geq 0$ too. In this case, we add the sign of boundary data point $\bar{x}_i$ to $S_{i,k}$ for future use in the second-order test. The operation with $S_{i,k}$ will be explained in detail in Appendix A.3. After checking if $g_k(z, \tilde{v}_k)^T \tilde{v}_k \geq 0$ holds for all extreme rays, FO-Increasing-Test returns False with $\{S_{i,k}\}_{i \in B_k}$.

**Counting flat extreme rays.** How many of these extreme rays satisfy $g_k(z, \tilde{v}_k)^T \tilde{v}_k = 0$? Presence of such flat extreme rays introduce inequality constraints in the QP that we will solve in SO-TEST (Algorithm 5). It is ideal not to have flat extreme rays, because in this case there are only equality constraints, so the QP is easier to solve. The following lemma shows conditions for existence of flat extreme rays as well as output of Algorithm 4.

**Lemma A.1.** *Suppose* $\mathbf{0}_{d_x+1}^T \in \mathcal{D}_k$ *and all extreme rays* $\tilde{v}_k$ *satisfy* $g_k(z, \tilde{v}_k)^T \tilde{v}_k \geq 0$. *Consider all* $i \in B_k$, *and its corresponding* $\hat{v}_{i,k} \in \mathcal{V}_k \cap \mathrm{span}\{\bar{x}_j \mid j \in B_k \setminus \{i\}\}^{\perp}$.

1. *If* $[W_2]_{\cdot,k}^T \nabla \ell_i = 0$, *then both extreme rays* $\hat{v}_{i,k}$ *and* $-\hat{v}_{i,k}$ *are flat extreme rays, and* $S_{i,k} = \{-1, +1\}$ *at the end of Algorithm 4.*

2. *If* $[W_2]_{\cdot,k}^T \nabla \ell_i \neq 0$ *and* $s_i^* = s_+$ *(or* $s_-$*), one (and only) of* $\tilde{v}_k \in \{\hat{v}_{i,k}, -\hat{v}_{i,k}\}$ *that satisfies* $\mathrm{sign}(\bar{x}_i^T \tilde{v}_k) = +1$ *(or* $-1$*) is a flat extreme ray, and* $S_{i,k} = \{+1\}$ *(or* $\{-1\}$*) at the end of Algorithm 4.*

3. *If* $[W_2]_{\cdot,k}^T \nabla \ell_i \neq 0$ *and* $s_i^* \neq s_{\pm}$, *both* $\hat{v}_{i,k}$ *and* $-\hat{v}_{i,k}$ *are not flat extreme rays, and* $S_{i,k} = \{0\}$ *at the end of Algorithm 4.*

**Proof** First note that we already assumed that all extreme rays $\tilde{v}_k$ satisfy $g_k(z, \tilde{v}_k)^T \tilde{v}_k \geq 0$, so SOSP-CHECK will reach Line 14 at the end. Also note that $\bar{x}_i$'s in $i \in B_k$ are linearly independent (by Lemma 1), so $\bar{x}_i^T \hat{v}_{i,k} \neq 0$.

If $[W_2]_{\cdot,k}^T \nabla \ell_i = 0$, then $(s_{\sigma_{i,k}} - s_i^*)[W_2]_{\cdot,k}^T \nabla \ell_i \bar{x}_i^T \tilde{v}_k = 0$ regardless of $\tilde{v}_k$, so both $\hat{v}_{i,k}$ and $-\hat{v}_{i,k}$ are flat extreme rays. If $[W_2]_{\cdot,k}^T \nabla \ell_i \neq 0$ and $s_i^* = s_+$, $\tilde{v}_k \in \{\hat{v}_{i,k}, -\hat{v}_{i,k}\}$ that satisfies $\mathrm{sign}(\bar{x}_i^T \tilde{v}_k) = +1$ gives $\sigma_{i,k} = +1$, so $s_{\sigma_{i,k}} = s_i^*$. Thus, $\tilde{v}_k$ is a flat extreme ray. The case with $s_i^* = s_-$ is proved similarly. If $[W_2]_{\cdot,k}^T \nabla \ell_i \neq 0$ and $s_i^* \neq s_+$, none of $(s_{\pm} - s_i^*)$, $[W_2]_{\cdot,k}^T \nabla \ell_i$, and $\bar{x}_i^T \hat{v}_{i,k}$ are zero, so $\hat{v}_{i,k}$ and $-\hat{v}_{i,k}$ cannot be flat. $\qquad\square$

Let $B_k^{(j)} \subseteq B_k$ denote the set of indices $i \in B_k$ satisfying conditions in Lemma A.1.$j$ ($j = 1, 2, 3$). Note that $B_k^{(j)}$'s partition the set $B_k$. We denote the union of $B_k^{(1)}$ and $B_k^{(2)}$ by $B_k^{(1,2)}$, and similarly, $B_k^{(2,3)} := B_k^{(2)} \cup B_k^{(3)}$. We can see from the lemma that $|S_{i,k}| = 2$ for $i \in B_k^{(1)}$, and $|S_{i,k}| = 1$ for $i \in B_k^{(2,3)}$. Also, it follows from the definition of $K$ and $L$ (5) that

$$K = \sum_{k=1}^{d_h} |B_k^{(1)}|, \quad L = \sum_{k=1}^{d_h} |B_k^{(1)}| + |B_k^{(2)}|.$$

**Connection to KKT conditions.** As a side remark, we provide connections of our tests to the well-known KKT conditions. Note that the equality $g_k(z, v_k)^T = [W_2]_{\cdot,k}^T \left( \sum_{i \in B_k} (s_{\sigma_i} - s_i^*) \nabla \ell_i \bar{x}_i^T \right)$ for $\sigma_i \bar{x}_i^T v_k \geq 0$, $\forall i \in B_k$ corresponds to the KKT stationarity condition, where $(s_{\sigma_i} - s_i^*)[W_2]_{\cdot,k}^T \nabla \ell_i$'s correspond to the Lagrange multipliers for inequality constraints. Then, testing extreme rays is equivalent to testing dual feasibility of Lagrange multipliers, and having zero dual variables ($[W_2]_{\cdot,k}^T \nabla \ell_i = 0$ or $s_i^* = s_+$ or $s_-$, resulting in flat extreme rays) corresponds to having degeneracy in the complementary slackness condition.

As mentioned in Section 2.1, given that $g(z, \eta)$ and $H(z, \eta)$ are constant functions of $\eta$ in each polyhedral cone, one can define inequality constrained optimization problems and try to solve for KKT conditions for $z$ directly. However, this also requires solving $2^M$ problems. The strength of our approach is that by solving the QPs (2), we can automatically compute the exact Lagrange multipliers for all $2^M$ subproblems, and dual feasibility is also tested in $O(M)$ time.

## A.3 Testing $\eta^T H(z, \eta)\eta \geq 0$ for $\{\eta \mid g(z, \eta)^T \eta = 0\}$ (Lines 19–32)

The second-order test checks $\eta^T H(z, \eta)\eta \geq 0$ for "flat" $\eta$'s satisfying $g(z, \eta)^T \eta = 0$. This is done with help of the function SO-TEST in Algorithm 5. Given its input $\{\sigma_{i,k}\}_{k \in [d_h], i \in B_k}$, it defines fixed "Jacobian" matrices $J_i$ for all data points and equality/inequality constraints for boundary data points, and solves the QP (4).

**Equality/inequality constraints.** In the QP (4), there are $d_h$ equality constraints of the form $[W_2]_{.,k}^T u_k = [[W_1]_{k,.} \quad [b_1]_k] v_k$. These equality constraints are due to the nonnegative homogeneous property of activation function $h$: scaling $[W_1]_{k,.}$ and $[b_1]_k$ by $\alpha > 0$ and scaling $[W_2]_{.,k}$ by $1/\alpha$ yields exactly the same network. This observation is stated more precisely in the following lemma.

**Lemma A.2.** *Suppose $z$ is a FOSP (differentiable or not) of $\mathfrak{R}(\cdot)$. Fix any $k \in [d_h]$, and define perturbation $\eta$ as*

$$u_k = -[W_2]_{.,k}, \ v_k = [[W_1]_{k,.} \quad [b_1]_k]^T, \ u_{k'} = \mathbf{0}, v_{k'} = \mathbf{0} \text{ for all } k' \neq k, \ \delta_2 = \mathbf{0}.$$

*Then, $g(z, \eta)^T \eta = \eta^T H(z, \eta) \eta = 0$.*

The proof of Lemma A.2 can be found in Appendix B.5. A corollary of this lemma is that any differentiable FOSP of $\mathfrak{R}$ always has rank-deficient Hessian, and the multiplicity of zero eigenvalue is at least $d_h$. Hence, these $d_h$ equality constraints on $u_k$'s and $v_k$'s force $\eta$ to be orthogonal to the loss-invariant directions.

The equality constraints of the form $\bar{x}_i^T v_k = 0$ are introduced when $\sigma_{i,k} = 0$; this happens for boundary data points $i \in B_k^{(3)}$. Therefore, there are $M - L$ additional equality constraints. The inequality constraints come from $i \in B_k^{(1,2)}$. So there are $L$ inequality constraints. Now, the following lemma proves that feasible sets defined by these equality/inequality constraints added to (4) exactly correspond to the regions where $g_k(z, v_k)^T v_k = 0$. Recall from Lemma A.1 that $S_{i,k} = \{-1, +1\}$ for $i \in B_k^{(1)}$, $S_{i,k} = \{-1\}$ or $\{+1\}$ for $i \in B_k^{(2)}$, and $S_{i,k} = \{0\}$ for $i \in B_k^{(3)}$.

**Lemma A.3.** *Let $\{\sigma_{i,k}\}_{i \in B_k^{(2)}}$ be the only element of $\prod_{i \in B_k^{(2)}} S_{i,k}$. Then, in SO-TEST,*

$$\bigcup_{\{\sigma_{i,k}\}_{i \in B_k^{(1)}} \in \prod_i S_{i,k}} \left\{ v_k \mid \forall i \in B_k^{(3)}, \bar{x}_i^T v_k = 0, \text{ and } \forall i \in B_k^{(1,2)}, \sigma_{i,k} \bar{x}_i^T v_k \geq 0 \right\}$$

$$= \left\{ v_k \mid \forall i \in B_k^{(3)}, \bar{x}_i^T v_k = 0, \text{ and } \forall i \in B_k^{(2)}, \sigma_{i,k} \bar{x}_i^T v_k \geq 0 \right\}$$

$$= \left\{ v_k \mid g_k(z, v_k)^T v_k = 0 \right\}.$$

The proof of Lemma A.3 is in Appendix B.6.

In total, there are $d_h + M - L$ equality constraints and $L$ inequality constraints in each nonconvex QP. It is also easy to check that these constraints are all linearly independent.

**How many QPs do we solve?** Note that in Line 19, we call SO-TEST with $\{\sigma_{i,k}\}_{k \in [d_h], i \in B_k} = \mathbf{0}$, which results in a QP (4) with $d_h + M$ equality constraints. This is done even when we have flat extreme rays, just to take a quick look if a descent direction can be obtained without having to deal with inequality constraints.

If there exist flat extreme rays (Line 22), the algorithm calls SO-TEST for each element of $\prod_{k \in [d_h]} \prod_{i \in B_k} S_{i,k}$. Recall that $|S_{i,k}| = 2$ for $i \in B_k^{(1)}$, so

$$\left| \prod_{k \in [d_h]} \prod_{i \in B_k} S_{i,k} \right| = 2^K.$$

In summary, if there is no flat extreme ray, the algorithm solves just one QP with $d_h + M$ equality constraints. If there are flat extreme rays, the algorithm solves one QP with $d_h + M$ equality constraints, and $2^K$ QPs with $d_h + M - L$ equality constraints and $L$ inequality constraints. This is also an improvement from the naive approach of solving $2^M$ QPs.

**Concluding the test.** After solving the QP, SO-TEST returns result to SOSP-CHECK. The algorithm returns two booleans and one perturbation tuple. The first is to indicate that there is no solution, i.e., there is a descent direction that leads to $-\infty$. Whenever there was any descent direction then we immediately return the direction and terminate. The second boolean is to indicate that there are nonzero $\eta$ that satisfies $\eta^T H(z, \eta) \eta = 0$. After solving all QPs, if any of SO-TEST calls found out $\eta \neq \mathbf{0}$ such that $g(z, \eta)^T \eta = 0$ and $\eta^T H(z, \eta) \eta = 0$, then we conclude SOSP-CHECK with "SOSP." If all QPs terminated with unique minimum at zero, then we can conclude "Local Minimum."

# B PROOF OF LEMMAS

## B.1 PROOF OF LEMMA 1

By definition, we have $[W_1]_{k,\cdot} x_i + [b_1]_k = 0$ for all $i \in B_k$, meaning that they are all on the same hyperplane $[W_1]_{k,\cdot} x + [b_1]_k = 0$. By the assumption, we cannot have more than $d_x$ points on the hyperplane.

Next, assume for the sake of contradiction that the $M_k := |B_k|$ data points $\bar{x}_i$'s are linearly dependent, i.e., there exists $a_1, \ldots, a_{M_k} \in \mathbb{R}$, not all zero, such that

$$\sum_{i=1}^{M_k} a_i \begin{bmatrix} x_i \\ 1 \end{bmatrix} = 0 \implies a_1 = -\sum_{i=2}^{M_k} a_i \implies \sum_{i=2}^{M_k} a_i (x_i - x_1) = 0,$$

where $a_2, \ldots, a_{M_k}$ are not all zero. This implies that these $M_k$ points $x_i$'s are on the same $(M_k - 2)$-dimensional affine space. To see why, consider for example the case $M_k = 3$: $a_2(x_2 - x_1) = -a_3(x_3 - x_1)$, meaning that they have to be on the same line. By adding any $d_x + 1 - M_k$ additional $x_i$'s, we can see that $d_x + 1$ points are on the same $(d_x - 1)$-dimensional affine space, i.e., a hyperplane in $\mathbb{R}^{d_x}$. This contradicts Assumption 2.

## B.2 PROOF OF LEMMA 2

From Assumption 1, $\ell(w, y)$ is twice differentiable and convex in $w$. By Taylor expansion of $\ell(\cdot)$ at $(Y(x_i), y_i)$,

$$\begin{aligned}
\mathfrak{R}(z + \eta) &= \sum_{i=1}^{m} \ell(Y(x_i) + dY(x_i), y_i) \\
&= \sum_{i=1}^{m} \ell(Y(x_i), y_i) + \nabla \ell_i^T dY(x_i) + \tfrac{1}{2} dY(x_i)^T \nabla^2 \ell_i dY(x_i) + o(\|\eta\|^2) \\
&= \mathfrak{R}(z) + \sum_{i=1}^{m} \nabla \ell_i^T dY_1(x_i) + \sum_{i=1}^{m} \nabla \ell_i^T dY_2(x_i) + \tfrac{1}{2} \sum_{i=1}^{m} \|dY_1(x_i)\|_{\nabla^2 \ell_i}^2 + o(\|\eta\|^2),
\end{aligned}$$

where the first-order term $\sum_{i=1}^{m} \nabla \ell_i^T dY_1(x_i) = \sum_{i=1}^{m} \nabla \ell_i^T (\Delta_2 O(x_i) + \delta_2 + W_2 J(x_i)(\Delta_1 x_i + \delta_1))$ can be further expanded to show

$$\sum_{i=1}^{m} \nabla \ell_i^T (\Delta_2 O(x_i) + \delta_2) = \left\langle \Delta_2, \sum_i \nabla \ell_i O(x_i)^T \right\rangle + \left\langle \delta_2, \sum_i \nabla \ell_i \right\rangle,$$

$$\sum_{i=1}^{m} \nabla \ell_i^T (W_2 J(x_i)(\Delta_1 x_i + \delta_1)) = \text{tr} \left( \sum_{i=1}^{m} J(x_i) W_2^T \nabla \ell_i \bar{x}_i^T \begin{bmatrix} \Delta_1^T \\ \delta_1^T \end{bmatrix} \right)$$

$$= \sum_{k=1}^{d_h} [W_2]_{\cdot,k}^T \left( \sum_{i=1}^{m} [J(x_i)]_{k,k} \nabla \ell_i \bar{x}_i^T \right) v_k = \sum_{k=1}^{d_h} [W_2]_{\cdot,k}^T \left( C_k + \sum_{i \in B_k} h'(\bar{x}_i^T v_k) \nabla \ell_i \bar{x}_i^T \right) v_k.$$

Also, note that in each of the $2^M$ divided region (which is a polyhedral cone) of $\eta$, $J(x_i)$ stays constant for all $i \in [m]$; thus, $g(z, \eta)$ and $H(z, \eta)$ are piece-wise constant functions of $\eta$. Specifically, since the parameter space is partitioned into polyhedral cones, we have $g(z, \eta) = g(z, \gamma\eta)$ and $H(z, \eta) = H(z, \gamma\eta)$ for any $\gamma > 0$.

## B.3 PROOF OF LEMMA 3

Suppose that $w_1, w_2, \ldots, w_q$ are orthonormal basis of $\text{row}(A)$. Choose $w_{q+1}, \ldots, w_p$ so that $w_1, w_2, \ldots, w_p$ form an orthonormal basis of $\mathbb{R}^p$. Let $W$ be an orthogonal matrix whose columns are $w_1, w_2, \ldots, w_p$, and $\hat{W}$ be an submatrix of $W$ whose columns are $w_{q+1}, \ldots, w_p$. With this definition, note that $I - A^T (AA^T)^{-1} A = \hat{W} \hat{W}^T$.

Suppose that we are given $\eta^{(t)}$ satisfying $A\eta^{(t)} = 0$. Then we can write $\eta^{(t)} = \hat{W}\mu^{(t)}$, where $\mu^{(t)} \in \mathbb{R}^{p-q}$ and $[\mu^{(t)}]_i = w_{i+q}^T \eta^{(t)}$. Define $\mu^{(t+1)}$ likewise. Then, noting $\eta^{(t)} = \hat{W}\hat{W}^T \eta^{(t)}$ gives

$$\hat{W}\mu^{(t+1)} = \eta^{(t+1)} = \eta^{(t)} - \alpha \hat{W}\hat{W}^T Q\hat{W}\mu^{(t)} = \hat{W}(I - \alpha \hat{W}^T Q\hat{W})\mu^{(t)}.$$

Define $C := \hat{W}^T Q \hat{W} \in \mathbb{R}^{(p-q)\times(p-q)}$, and then write its eigen-decomposition $C = VSV^T$ and denote its eigenvectors as $\nu_1, \dots, \nu_{p-q}$ and its corresponding eigenvalues $\lambda_1, \dots, \lambda_{p-q}$. Then note

$$\mu^{(t+1)} = (I - \alpha C)\mu^{(t)} = (I - \alpha VSV^T)\sum_{i=1}^{p-q}(\nu_i^T\mu^{(t)})\nu_i = \sum_{i=1}^{p-q}(1-\alpha\lambda_i)(\nu_i^T\mu^{(t)})\nu_i$$

$$= \sum_{i=1}^{p-q}(1-\alpha\lambda_i)^2(\nu_i^T\mu^{(t-1)})\nu_i = \cdots = \sum_{i=1}^{p-q}(1-\alpha\lambda_i)^{t+1}(\nu_i^T\mu^{(0)})\nu_i.$$

This proves that this iteration converges or diverges exponentially fast. Starting from the initial point $\eta^{(0)} = \hat{W}\mu^{(0)}$, the component of $\mu^{(0)}$ that corresponds to negative eigenvalue blows up exponentially fast, those corresponding to positive eigenvalue shrinks to zero exponentially fast (if $\alpha < 1/\lambda_{\max}(C)$), and those with zero eigenvalue will stay invariant. Therefore, if there exists $\lambda_i < 0$, then $\eta^{(t)}$ blows up to infinity quickly and finds an $\eta$ such that $\eta^T Q \eta < 0$ (T3). If all $\lambda_i \geq 0$, it converges exponentially fast to $\hat{W}\sum_{i:\lambda_i=0}(\nu_i^T\mu^{(0)})\nu_i$ (T2). If all $\lambda_i > 0$, $\eta^{(t)} \to \mathbf{0}$ (T1).

It is left to prove that $\alpha < 1/\lambda_{\max}(Q)$ guarantees convergence, as stated. To this end, it suffices to show that $\lambda_{\max}(Q) \geq \lambda_{\max}(C)$. Note that

$$C = \hat{W}^T Q \hat{W} = \hat{W}^T W W^T Q W W^T \hat{W} = [\mathbf{0} \quad I] W^T Q W \begin{bmatrix}\mathbf{0}\\I\end{bmatrix}.$$

Using the facts that $\lambda_{\max}(Q) = \lambda_{\max}(W^T Q W)$ and $C$ is a principal submatrix of $W^T Q W$,

$$\lambda_{\max}(Q) = \max_x \frac{x^T W^T Q W x}{x^T x} \geq \max_{x:[x]_{1:q}=\mathbf{0}} \frac{x^T W^T Q W x}{x^T x} = \lambda_{\max}(C).$$

Also, if we start at a random initial point (e.g., sample from a Gaussian in $\mathbb{R}^p$ and project to $\text{row}(A)^\perp$), then with probability 1 we have $\nu_i^T\mu^{(0)} \neq 0$ for all $i \in [p-q]$, so we will get the correct convergence/divergence result almost surely.

## B.4 PROOF OF LEMMA 4

### B.4.1 PRELIMINARIES

Before we prove the complexity lemma, we introduce the definitions of copositivity and Pareto spectrum, which are closely related concepts to our specific form of QP.

**Definition B.1.** *Let $Q \in \mathbb{R}^{r\times r}$ be a symmetric matrix. We say that $Q$ is copositive if $\eta^T Q \eta \geq 0$ for all $\eta \geq \mathbf{0}$. Moreover, strict copositivity means that $\eta^T Q \eta > 0$ for all $\eta \geq \mathbf{0}$, $\eta \neq \mathbf{0}$.*

Testing whether $Q$ is not copositive known to be NP-complete (Murty & Kabadi, 1987); it is certainly a difficult problem. There is a method testing cositivity of $Q$ in $O(r^3 2^r)$ time which uses its Pareto spectrum $\Pi(Q)$. The following is the definition of Pareto spectrum, taken from Seeger (1999); Hiriart-Urruty & Seeger (2010).

**Definition B.2.** *Consider the problem*

$$\underset{\eta \geq \mathbf{0}, \|\eta\|_2 = 1}{\text{minimize}} \quad \eta^T Q \eta.$$

*KKT conditions for the above problem gives us a complementarity system*

$$\eta \geq \mathbf{0}, \quad Q\eta - \lambda\eta \geq \mathbf{0}, \quad \eta^T(Q\eta - \lambda\eta) = 0, \|\eta\|_2 = 1, \tag{6}$$

*where $\lambda \in \mathbb{R}$ is viewed as a Lagrange multiplier associated with $\|\eta\|_2 = 1$. The number $\lambda \in \mathbb{R}$ is called a Pareto eigenvalue of $Q$ if (6) admits a solution $\eta$. The set of all Pareto eigenvalues of $Q$, denoted as $\Pi(Q)$, is called the Pareto spectrum of $Q$.*

The next lemma reveals the relation of copositivity and Pareto spectrum:

**Lemma B.1** (Theorem 4.3 of Hiriart-Urruty & Seeger (2010)). *A symmetric matrix $Q$ is copositive (or strictly copositive) if and only if all the Pareto eigenvalues of $Q$ are nonnegative (or strictly positive).*

Now, the following lemma tells us how to compute Pareto spectrum of $Q$.

**Lemma B.2** (Theorem 4.1 of Seeger (1999)). *Let $Q$ be a matrix of order $r$. Consider a nonempty index set $J \subseteq [r]$. Given $J$, $Q^J$ refers to the principal submatrix of $Q$ with the rows and columns of $Q$ indexed by $J$. Let $2^{[r]} \setminus \emptyset$ denote the set of all nonempty subsets of $[r]$. Then $\lambda \in \Pi(Q)$ if and only if there exists an index set $J \in 2^{[r]} \setminus \emptyset$ and a vector $\xi \in \mathbb{R}^{|J|}$ such that*

$$Q^J \xi = \lambda \xi, \ \ \xi \in \text{int}(\mathbb{R}_+^{|J|}), \ \ \sum_{j \in J} [Q]_{i,j} [\xi]_j \geq 0 \text{ for all } i \notin J.$$

*In such a case, the vector $\eta \in \mathbb{R}^r$ by*

$$[\eta]_j = \begin{cases} [\xi]_j & \text{if } j \in J, \\ 0 & \text{if } j \notin J \end{cases}$$

*is a Pareto-eigenvector of $Q$ associated to the Pareto eigenvalue $\lambda$.*

These lemmas tell us that the Pareto spectrum of $Q$ can be calculated by computing eigensystems of all $2^r - 1$ possible $Q^J$, which takes $O(r^3 2^r)$ time in total, and from this we can determine whether a symmetric $Q$ is copositive.

### B.4.2 PROOF OF THE LEMMA

With the preliminary concepts presented, we now start proving our Lemma 4. We will first transform $\eta$ to eliminate the equality constraints and obtain an inequality constrained problem of the form $\text{minimize}_{w:\bar{B}w \geq 0} w^T R w$. From there, we can use the theorems from Martin & Jacobson (1981), which tell us that by testing positive definiteness of a $(p-q-r) \times (p-q-r)$ matrix and copositivity of a $r \times r$ matrix we can determine which of the three categories the QP falls into. Transforming $\eta$ and testing positive definiteness take $O(p^3)$ time and testing copositivity takes $O(r^3 2^r)$ time, so the test in total is done in $O(p^3 + r^3 2^r)$ time.

We now describe how to transform $\eta$ and get an equivalent optimization problem of the form we want. We assume without loss of generality that $A = [A_1 \quad A_2]$ where $A_1 \in \mathbb{R}^{q \times q}$ is invertible. If not, we can permute components of $\eta$. Then make a change of variables

$$\eta = T_A \begin{bmatrix} \bar{w} \\ w \end{bmatrix} := \begin{bmatrix} A_1^{-1} & -A_1^{-1} A_2 \\ \mathbf{0}_{(p-q) \times q} & I_{p-q} \end{bmatrix} \begin{bmatrix} \bar{w} \\ w \end{bmatrix}, \text{ so that } A T_A \begin{bmatrix} \bar{w} \\ w \end{bmatrix} = [I \quad \mathbf{0}] \begin{bmatrix} \bar{w} \\ w \end{bmatrix} = \bar{w}.$$

Consequently, the constraint $A\eta = 0$ becomes $\bar{w} = 0$. Now partition $B = [B_1 \quad B_2]$, where $B_1 \in \mathbb{R}^{r \times q}$. Also let $R$ be the principal submatrix of $T_A^T Q T_A$ composed with the last $p - q$ rows and columns. It is easy to check that

$$\begin{array}{ll} \text{minimize}_\eta & \eta^T Q \eta \\ \text{subject to} & A\eta = \mathbf{0}_q, \ B\eta \geq \mathbf{0}_r. \end{array} \quad \equiv \quad \begin{array}{ll} \text{minimize}_w & w^T R w \\ \text{subject to} & (B_2 - B_1 A_1^{-1} A_2) w \geq \mathbf{0}_r. \end{array}$$

Let us quickly check if $B_2 - B_1 A_1^{-1} A_2$ has full row rank. One can observe that

$$\begin{bmatrix} A_1 & A_2 \\ B_1 & B_2 \end{bmatrix} = \begin{bmatrix} I_q & \mathbf{0} \\ B_1 A_1^{-1} & I_r \end{bmatrix} \begin{bmatrix} A_1 & A_2 \\ \mathbf{0} & B_2 - B_1 A_1^{-1} A_2 \end{bmatrix}.$$

It follows from the assumption $\text{rank}([A^T \quad B^T]) = q + r$ that $\bar{B} := B_2 - B_1 A_1^{-1} A_2$ has rank $r$, which means it has full row rank.

Before stating the results from Martin & Jacobson (1981), we will transform the problem a bit further. Again, assume without loss of generality that $\bar{B} = [\bar{B}_1 \quad \bar{B}_2]$ where $\bar{B}_1 \in \mathbb{R}^{r \times r}$ is invertible. Define another change of variables as the following:

$$w = T_B \nu := \begin{bmatrix} \bar{B}_1^{-1} & -\bar{B}_1^{-1} \bar{B}_2 \\ \mathbf{0}_{(p-q-r) \times r} & I_{p-q-r} \end{bmatrix} \begin{bmatrix} \nu_1 \\ \nu_2 \end{bmatrix}, \ \ T_B^T R T_B =: \begin{bmatrix} \bar{R}_{11} & \bar{R}_{12} \\ \bar{R}_{12}^T & \bar{R}_{22} \end{bmatrix} =: \bar{R}.$$

Consequently, we get

$$\begin{array}{ll} \text{minimize}_w & w^T R w \\ \text{subject to} & \bar{B} w \geq \mathbf{0}_r. \end{array} \quad \equiv \quad \begin{array}{ll} \text{minimize}_w & \nu^T \bar{R} \nu = \nu_1^T \bar{R}_{11} \nu_1 + 2 \nu_1^T \bar{R}_{12} \nu_2 + \nu_2^T \bar{R}_{22} \nu_2 \\ \text{subject to} & \nu_1 \geq \mathbf{0}_r. \end{array}$$

Given this transformation, we are ready to state the lemmas.

**Lemma B.3** (Theorem 2.2 of Martin & Jacobson (1981)). *If $\bar{B} = \begin{bmatrix} \bar{B}_1 & \bar{B}_2 \end{bmatrix}$, with $\bar{B}_1$ $r \times r$ invertible, then with $\bar{R}_{ij}$'s given as above, $w^T R w > 0$ whenever $\bar{B} w \geq \mathbf{0}$, $w \neq \mathbf{0}$ if and only if*

- *$\bar{R}_{22}$ is positive definite, and*
- *$\bar{R}_{11} - \bar{R}_{12} \bar{R}_{22}^{-1} \bar{R}_{12}^T$ is strictly copositive.*

**Lemma B.4** (Theorem 2.1 of Martin & Jacobson (1981)). *If $\bar{B} = \begin{bmatrix} \bar{B}_1 & \bar{B}_2 \end{bmatrix}$, with $\bar{B}_1$ $r \times r$ invertible, then with $\bar{R}_{ij}$'s given as above, $w^T R w \geq 0$ whenever $\bar{B} w \geq \mathbf{0}$ if and only if*

- *$\bar{R}_{22}$ is positive semidefinite, $\mathrm{null}(\bar{R}_{22}) \subseteq \mathrm{null}(\bar{R}_{12})$, and*
- *$\bar{R}_{11} - \bar{R}_{12} \bar{R}_{22}^\dagger \bar{R}_{12}^T$ is copositive,*

*where $\bar{R}_{22}^\dagger$ is a pseudoinverse of $\bar{R}_{22}$.*

Using Lemmas B.3 and B.4, we now describe how to test our given QP and declare one of (T1), (T2), or (T3). First, we compute the eigensystem of $\bar{R}_{22}$ and see which of the following disjoint categories it belongs to:

(PD1) All eigenvalues $\lambda_1, \ldots, \lambda_{p-q-r}$ of $\bar{R}_{22}$ satisfy $\lambda_i > 0$.

(PD2) $\forall i, \lambda_i \geq 0$, but $\exists i$ such that $\lambda_i = 0$, and $\forall \nu_2$ s.t. $\bar{R}_{22} \nu_2 = 0$, we have $\bar{R}_{12} \nu_2 = 0$.

(PD3) $\forall i, \lambda_i \geq 0$, but $\exists i$ such that $\lambda_i = 0$, and $\exists \nu_2$ s.t. $\bar{R}_{22} \nu_2 = 0$ but $\bar{R}_{12} \nu_2 \neq 0$.

(PD4) $\exists i$ such that $\lambda_i < 0$, i.e., $\exists \nu_2$ such that $\nu_2^T \bar{R}_{22} \nu_2 < 0$.

If the test comes out (PD3) or (PD4), then we can immediately declare (T3) without having to look at copositivity. This is because if we get (PD4), we can choose $\nu_1 = 0$ so that $\nu^T \bar{R} \nu = \nu_2^T \bar{R}_{22} \nu_2 < 0$. In case of (PD3), one can fix any $\nu_1$ satisfying $\nu_1^T \bar{R}_{12} \nu_2 \neq 0$, and by scaling $\nu_2$ to positive or negative we can get $\nu^T \bar{R} \nu \to -\infty$. Notice that once we have these $\nu$ satisfying $\nu^T \bar{R} \nu < 0$, we can recover $\eta$ from $\nu$ by backtracking the transformations.

Next, compute the Pareto spectrum of $S := \bar{R}_{11} - \bar{R}_{12} \bar{R}_{22}^\dagger \bar{R}_{12}^T$ and check which case $S$ belongs to:

(CP1) $S = \bar{R}_{11} - \bar{R}_{12} \bar{R}_{22}^\dagger \bar{R}_{12}^T$ is strictly copositive.

(CP2) $S$ is copositive, but $\exists \nu_1 \geq \mathbf{0}, \nu_1 \neq \mathbf{0}$ such that $\nu_1^T S \nu_1 = 0$.

(CP3) $\exists \nu_1 \geq \mathbf{0}$ such that $\nu_1^T S \nu_1 < 0$.

Here, $\nu_1$'s are Pareto eigenvectors of $S$ defined in Lemma B.2. If we have (CP3), we can declare (T3) because one can fix $\nu_2 = -\bar{R}_{22}^\dagger R_{12}^T \nu_1$ and get $\nu^T \bar{R} \nu = \nu_1^T S \nu_1 < 0$. If the tests come out (PD1) and (CP1), by Lemma B.3 we have (T1). For the remaining cases, we conclude (T2).

## B.5 PROOF OF LEMMA A.2

With the given $\eta$,

$$\Delta_1 = \begin{bmatrix} \mathbf{0}_{(k-1) \times d_x} \\ [W_1]_{k,\cdot} \\ \mathbf{0}_{(d_h-k) \times d_x} \end{bmatrix}, \quad \delta_1 = \begin{bmatrix} \mathbf{0}_{k-1} \\ [b_1]_k \\ \mathbf{0}_{d_h-k} \end{bmatrix}, \quad \Delta_2 = \begin{bmatrix} \mathbf{0}_{d_y \times (k-1)} & -[W_2]_{\cdot,k} & \mathbf{0}_{d_y \times (d_h-k)} \end{bmatrix}.$$

It is straightforward to check that for all $i \in [m]$,

$$dY_1(x_i) = \Delta_2 O(x_i) + W_2 J(x_i)(\Delta_1 x_i + \delta_1) = -[O(x_i)]_k [W_2]_{\cdot,k} + W_2 \begin{bmatrix} \mathbf{0}_{k-1} \\ [O(x_i)]_k \\ \mathbf{0}_{d_h-k} \end{bmatrix} = \mathbf{0}.$$

From this, $g(z, \eta)^T \eta = \sum_i \nabla \ell_i^T dY_1(x_i) = 0$. For the second order terms,

$$\eta^T H(z, \eta) \eta = \sum_{i=1}^m \nabla \ell_i^T dY_2(x_i) + \tfrac{1}{2} \sum_{i=1}^m \|dY_1(x_i)\|_{\nabla^2 \ell_i}^2 = \sum_{i=1}^m \nabla \ell_i^T \Delta_2 J(x_i)(\Delta_1 x_i + \delta_1)$$

$$= \sum_{i=1}^m \nabla \ell_i^T (-[O(x_i)]_k [W_2]_{\cdot,k}) = - \left( \sum_{i=1}^m [O(x_i)]_k \nabla \ell_i^T \right) [W_2]_{\cdot,k}.$$

From the fact that $z$ is a FOSP of $\mathfrak{R}$, it follows that $\sum_i \nabla \ell_i O(x_i)^T = \mathbf{0}$, so $\eta^T H(z, \eta) \eta = 0$.

### B.6 PROOF OF LEMMA A.3

The first equality is straightforward, because it follows from $S_{i,k} = \{-1, +1\}$ for all $i \in B_k^{(1)}$ that taking union of $\{\bar{x}_i^T v_k \leq 0\}$ and $\{\bar{x}_i^T v_k \geq 0\}$ will eliminate the inequality constraints for $i \in B_k^{(1)}$.

For the next equality, we start by expressing $\mathcal{U}_1 := \{v_k \mid g_k(z, v_k)^T v_k = 0\}$ as a linear combination of its linearly independent components. The set $\mathcal{U}_1$ can be expressed in the following form:

$$\mathcal{U}_1 = \{v_\perp + \sum_{i \in B_k^{(1)}} \alpha_i \hat{v}_{i,k} + \sum_{i \in B_k^{(2)}} \beta_i \hat{v}_{i,k} \mid v_\perp \in \mathcal{V}_k^\perp, \forall i \in B_k^{(1)}, \alpha_i \in \mathbb{R}, \text{ and } \forall i \in B_k^{(2)}, \beta_i \geq 0\},$$

where $\hat{v}_{i,k} \in \mathcal{V}_k \cap \text{span}\{\bar{x}_j \mid j \in B_k \setminus \{i\}\}^\perp$ for all $i \in B_k^{(1,2)}$. Additionally, for $i \in B_k^{(2)}$, $\hat{v}_{i,k}$ is in the direction that satisfies $\sigma_{i,k} = \text{sign}(\bar{x}_i^T \hat{v}_{i,k})$. To see why $\mathcal{U}_1$ can be expressed in such a form, first note that at the moment SO-TEST is executed, it is already given that the point $z$ is a FOSP. So, for any perturbation $v_k$ we have $g_k(z, v_k) \in \mathcal{V}_k$, and $g_k(z, v_k)^T v_\perp = 0$ for any $v_\perp \in \mathcal{V}_k^\perp$. For the remaining components, please recall FO-INCREASING-TEST and Lemma A.1; $\hat{v}_{i,k}$ are flat extreme rays, so they are the ones satisfying $g_k(z, v_k)^T v_k = 0$.

It remains to show that $\mathcal{U}_2 := \{v_k \mid \forall i \in B_k^{(3)}, \bar{x}_i^T v_k = 0, \text{ and } \forall i \in B_k^{(2)}, \sigma_{i,k} \bar{x}_i^T v_k \geq 0\} = \mathcal{U}_1$. We show this by proving $\mathcal{U}_1 \subseteq \mathcal{U}_2$ and $\mathcal{U}_1^c \subseteq \mathcal{U}_2^c$.

To show the first part, we start by noting that for any $v_\perp \in \mathcal{V}_k^\perp$, $\bar{x}_i^T v_\perp = 0$ for $i \in B_k^{(2,3)}$ because $\bar{x}_i \in \mathcal{V}_k$ for these $i$'s. Also, for all $i \in B_k^{(1)}$, it follows from the definition of $\hat{v}_{i,k}$ that $\bar{x}_j^T \hat{v}_{i,k} = 0$ for all $j \in B_k^{(2,3)}$. Similarly, for all $i \in B_k^{(2)}$, $\bar{x}_j^T \hat{v}_{i,k} = 0$ for all $j \in B_k^{(2,3)} \setminus \{i\}$, and $\sigma_{i,k} \bar{x}_i^T \hat{v}_{i,k} > 0$. Therefore, any $v_k \in \mathcal{U}_1$ must satisfy all constraints in $\mathcal{U}_2$, hence $\mathcal{U}_1 \subseteq \mathcal{U}_2$.

For the next part, we prove that $v_k \in \mathcal{U}_1^c$ violates at least one constraint in $\mathcal{U}_2$. Observe that the whole vector space $\mathbb{R}^p$ can be expressed as

$$\mathbb{R}^p = \{v_\perp + \sum_{i \in B_k^{(1)}} \alpha_i \hat{v}_{i,k} + \sum_{i \in B_k^{(2)}} \beta_i \hat{v}_{i,k} + w \mid w \in \mathcal{V}_k \cap \text{span}\{\hat{v}_{i,k}, i \in B_k^{(1,2)}\}^\perp,$$
$$v_\perp \in \mathcal{V}_k^\perp, \forall i \in B_k^{(1)}, \alpha_i \in \mathbb{R}, \text{ and } \forall i \in B_k^{(2)}, \beta_i \in \mathbb{R}\}.$$

Therefore, any $v_k \in \mathcal{U}_1^c$ either has a nonzero component $w$ in $V_k \cap \text{span}\{\hat{v}_{i,k}, i \in B_k^{(1,2)}\}^\perp$ or there exists $i \in B_2^{(k)}$ such that $\beta_i < 0$. By definition, $\hat{v}_{i,k} \in \text{span}\{\bar{x}_j \mid j \in B_k^{(3)}\}^\perp$ for any $i \in B_k^{(1,2)}$, which implies that $V_k \cap \text{span}\{\hat{v}_{i,k}, i \in B_k^{(1,2)}\}^\perp = \text{span}\{\bar{x}_j \mid j \in B_k^{(3)}\}$. Thus, a nonzero component $w \in \text{span}\{\bar{x}_j \mid j \in B_k^{(3)}\}$ will violate some equality constraints in $\mathcal{U}_2$. Next, in case where $\exists i \in B_2^{(k)}$ such that $\beta_i < 0$, this violates the inequality constraint corresponding to $i$.

