# OpenReview forum: "Efficiently testing local optimality and escaping saddles for ReLU networks"
_ICLR.cc/2019/Conference_

### Official Review · AnonReviewer3 · 2018-11-02
**Refreshing ideas, clever algorithm, unclear impact**

**Rating:** 8
**Confidence:** 3

**Review:**

The paper proposes a method to check if a given point is a stationary point or not (if not, it provides a descent direction), and then classify stationary points as either local min or second-order stationary. The method works for a specific non-differentiable loss.  In the worst case, there can be exponentially many flat directions to check (2^L), but usually this is no the case.

Overall, I'm impressed. The analysis seems solid, and a lot of clever ideas are used to get around issues (such as exponential number of regions, and non-convex QPs that cannot be solved by the S-procedure or simple tricks). A wide-variety of techniques are used: non-smooth analysis, recent analysis of non-convex QPs, copositive optimization.

The writing is clear and makes most arguments easy to follow.

There are some limitations:

(1) the technical details are hard to follow, and most are in a lengthy appendix, which I did not check

(2) there was no discussion of robustness. If I find a direction eta for which the directional derivative is zero, what do you mean by "zero"? This is implemented on a computer, so we don't really expect to find a directional derivative that is exactly zero.  I would have liked to see some discussions with epsilons, and give me a guarantee of an epsilon-SOSP or some kind of notion.  In the experiments, this isn't discussed (though another issue is touched on a little bit: you wanted to find real stationary points to test, but you don't have exactly stationary points, but rather can get arbitrarily close).  To make this practical, I think you need a robust theory.

(3) The numerical simulations mainly provided some evidence that there are usually not too many flat directions, but don't convince us that this is a useful technique on a real problem.  The discussion about possible loss functions at the end was a bit opaque.  Furthermore, if you can't find a dataset/loss, then why is this technique useful?

The paper is interesting and novel enough that despite the limitations, I am supportive of publishing it. It introduces new ideas that I find refreshing. The technique many not ever make it into the state-of-the-art algorithms, but I think the paper has intellectual value regardless of practical value.

In short, quality = high, clarity=high, originality=very high, and significance=hard-to-predict

---

> ### Author Response · Authors · 2018-11-22
> **Response to AnonReviewer3**
>
> Thank you very much for your feedback. We are glad that you enjoyed reading our paper. We list our answers to your comments, by their numbering:
>
> (1) Yes, we agree that there is certainly room for improvement; we will make our best efforts in revising the paper accordingly.
>
> (2) For the discussion on the robustness of the algorithm in general, we wrote a separate comment above to address common concerns raised by the reviewers. Please refer to item (1) of the general comments.
> Regarding the specific concern of testing if a directional derivative is zero, we believe that the reviewer is talking about testing the existence of flat extreme rays. In our experiments, to count the number of “approximate” flat extreme rays, we used our lemma A.1 that gives conditions for existence of flat extreme rays, and tested if these conditions are approximately satisfied. For more details, please refer to the end of the 2nd paragraph of Section 4.
>
> (3) The main purpose of our numerical experiments is to provide an empirical evidence of how many boundary data points (M) and flat extreme rays (L) we can have, because these quantities are difficult to estimate/bound theoretically. Our experiments show that, in our settings, there can be nonsmooth local minima with large M (implying that our analysis on nonsmooth points is meaningful) but L is usually surprisingly small.

---

> > ### Comment · AnonReviewer3 · 2018-11-26
> > **Response acknowledged**
> >
> > I read the authors' response. Given the shared concerns by all reviewers about robustness and applicability, I am not quite as positive as I was before, but I still support the paper.  The authors seem well-aware of the shortcomings of the work, which would require a major new work to address. I think this work is an interesting stepping-stone, and shows original thought.

---

> > > ### Author Response · Authors · 2018-11-28
> > > **Thank you for your feedback**
> > >
> > > Thank you for your positive feedback and support! As you pointed out, analyzing and implementing a robust version of the algorithm will require a significant amount of additional effort. We hope to tackle this in the future.

---

### Official Review · AnonReviewer2 · 2018-11-04
**Algorithm for testing local optimality in ReLU networks**

**Rating:** 6
**Confidence:** 3

**Review:**

Summary:
This work proposes a theoretical algorithm for checking local optimality and escaping saddles when training two-layer ReLU networks. The proposed "checking algorithm" involves solving convex and non-convex quadratic programs (QP) which can be done in polynomial time. The paper is well organized and technically correct with detailed proofs.

Comments:
1) Applicability issue: the conditions required by the proposed checking algorithm are too ideal, making it difficult to apply in practical applications. For example, the first step of the proposed algorithm is to check whether 0 belongs to the subdifferential. In practice, the iterates may get very close to a stationary point, but arriving to a stationary point might be too time-consuming and unrealistic. If the problem is smooth, then the gradient is expected to be small so that one can easily relax this first order optimality condition by allowing a small gradient. However, since here the problem is nonsmooth, in general the subgradient could be still very large even when the iterate is very close to a stationary point.  Therefore, one would need to relax the ideal conditions in the proposed algorithm to make it more applicable.

2) Another concern is that the efficiency of the proposed method relies too much on the empirical result that the number of flat extreme ray is small. The computational complexities for the test of the local optimality is exponentially depending on the number of flat extreme rays. Thus to guarantee a high efficiency of the proposed test algorithm and to make the main theory sound, it is important to provide a theoretical bound on this number. Without appropriate theoretical guarantees on the upper-bound of this number, it is not persuasive to claim that the proposed theoretical algorithm is of high efficiency.

3) The computational complexity is proportional to the number of training data points which could be huge. Is it possible to have a stochastic version?

Typos:
1) On page 2, under Section 2, ``$h(t):=$" should be ``$h(x):=$"

2) In section 2.1, at the end of the paragraph "Bisection by boundary data points": change $b_1$ by $\delta_1$ in ``$\Delta_1x_i+b_1$".

3) On page 4, when defining B_k, change x by x_i.

4) On page 5, above Lemma 1, when defining C_k, N(x_i) is not well defined.

---

> ### Author Response · Authors · 2018-11-22
> **Response to AnonReviewer2**
>
> We thank the reviewer for the time and effort invested in reviewing our paper. Below, we will address the comments point by point:
>
> 1) For the discussion on ideal conditions of the algorithm, we wrote a separate comment above to address common concerns raised by the reviewers. Please refer to item (1) of the comment.
>
> 2) We agree that we do not have a good theoretical bound on L, so in the worst case we might suffer exponential running time. Due to the complex nature of the loss surface of empirical risk minimization problems, providing tight theoretical bound for L might be very difficult, so we instead provide some empirical evidence showing L is usually small. We leave the theory side as future work.
>
> 3) Indeed, the computational cost of calculating exact (sub)differentials and Hessians grow proportionally with the number of data points m. It seems difficult to obtain a stochastic version unless we add assumptions on the distribution of data points. If we can develop a robust version of the algorithm as mentioned in item 1), then with some distributional assumptions on data, we expect that we can get some high probability results for a stochastic version.
> However, even without the stochastic version, we expect that (a numerical implementation of) our algorithm will be used only for testing local optimality almost at the end of training, not every iteration. Thus, its computational cost will not be too big.
>
> Thank you very much for pointing out those typos. $[N(x_i)]_k$ is originally meant to be $[W_1 x_i + b_1]_k$. We will fix these typos in the next revision.

---

### Official Review · AnonReviewer1 · 2018-11-05

**Rating:** 6
**Confidence:** 2

**Review:**

This paper proposes an efficient method to test whether a point is a local minimum in a 1-hidden-layer ReLU network. If the point is not a local minimum, the algorithm also returns a direction for descending the value of the loss function.

The tests include a first-order stationary point test (FOSP), and a second-order stationary point test (SOSP). As these test can be written as QPs, the core challenge is that if there are M boundary points in the dataset, i.e.,  data points on a non-differentiable region of the ReLU function, then the FOSP test requires 2^M tests of extreme rays -- each boundary partition the whole space into at least two parts. This paper observes that since the feasible sets are pointed polyhedral cones. Therefore checking only these extreme rays suffices. This results in an efficient test with only 2M tests.

Lastly, the paper performs experiments on synthetic data. It turns out there are surprisingly many boundary points.

Comments:
This paper proposes an interesting method of testing whether a given point is a local minimum or not in a ReLU network. The technique is non-trivial and requires some key observation to make it computationally efficient. However, I have the following concerns:
1) such a test may need very high numeric precision. For instance, you cannot make sure whether a floating point number is strictly greater than 0 or not. The small error may critically affect the property of a point.
2) boundary points of a ReLU network should have measure 0 (correct me if not). The finding in the experiment shows surprisingly many boundary points. This is counter-intuitive. Is it because of numeric issues? You might misclassify non-boundary points.
3) Usefulness.
    a. The paper claims that such a test would be very useful in practice. However, they cannot even perform an experiment on real datasets.
    b. Such a method only works for one-hidden layer network. It is not clear deeper network admit similar structure.
    c. Practical training of neural-network usually trains the network using SGD, which always obtain a solution with a non-zero gradient. In this sense, there is no need for such a testing.
    d. It seems like it is much easier to perform a test with different activation function, e.g., sigmoid.

If the authors can address these concerns convincingly, I would be happy to change the rating.

---

> ### Author Response · Authors · 2018-11-22
> **Response to AnonReviewer1**
>
> We appreciate the reviewer for their time and thoughtful comments. Below, we will provide answers to the reviewer’s concerns.
>
> 1) For the discussion on the precision of the algorithm, we wrote a separate comment above to address common concerns raised by the reviewers. Please refer to item (1) of the comment.
>
> 2) It is true that the set of nondifferentiable points has measure zero. On the other hand, please note that a nondifferentiable point can have multiple boundary data points, i.e., x_i’s that satisfy [W_1 x_i + b_1]_k = 0 for some k (input to the k-th hidden node is zero for this x_i). Also, such nondifferentiable points with many boundary data points lie on the intersection of subsets of the parameter space, where each subset corresponds to one boundary data point x_i and contains the parameter values satisfying [W_1 x_i + b_1]_k = 0.
> Our experiments were run for an extended period of time with exponentially decaying step size, to get as close to the exact nondifferentiable point (potentially local minimum) as possible. And then, we counted the number of “approximate” boundary data points, i.e., x_i’s that satisfied abs( [W_1 x_i + b_1]_k ) < 1e-5 for some k. In our experimental settings, it turns out that gradient descent pushes the parameters to a point with multiple boundary data points (i.e., M is large), but there are usually very few flat extreme rays (i.e., L is small).
>
> 3a) At our current stage of results, we are not claiming that our algorithm is useful in practice. By the experimental results, we are claiming the following: 1) given that M can be large in practice, our analysis of nondifferentiable points is meaningful, 2) L is usually very small in our experiments, so testing local minimality at nondifferentiable points can be tractable.
>
> 3b) Our analysis for now is limited to one-hidden-layer networks. For deeper networks, perturbation on the first layer may affect later layers, so the extension to deep networks is beyond the scope of this paper. For now, we leave this extension as future work.
>
> 3c) Due to multiple reviewers with similar concerns, we addressed this issue in a separate comment above. Please refer to item (2) of the comment.
>
> 3d) Your observation is true, because other activation functions do not have nondifferentiable points. In such cases, we can directly compute the gradient and Hessian, so the second-order stationarity test is straightforward. However, ReLU is one of the most popular activation functions, and it inevitably introduces nondifferentiable points in the empirical risk, which are difficult to analyze. The goal of our paper is to shed light on a better understanding of such nonsmooth points.

---

### Official Review · AnonReviewer4 · 2018-11-09

**Rating:** 3
**Confidence:** 4

**Review:**

Updates:
Author(s) acknowledged that they cannot get a robust analysis. Furthermore, the optimality test also requires a robust analysis. Therefore, I believe the current version is still incomplete so I changed my score. I encourage author(s) to add the robust analysis and submit to the next top machine learning conference.

-------------------------------------------
Paper Summary:
This paper gives a new algorithm to check whether a given point is a (generalized) second-order stationary point if not, it can return a strict descent direction even at this point the objective function (empirical risks of two-layer ReLU or Leaky-ReLU networks) is not differentiable.
The main challenge comes from the non-differentiability of ReLU. While testing a second-order stationary point is easy, because of the non-differentiability, one needs to test 2^M regions in the ReLU case. This paper exploits the special structure of two-layer ReLU network and shows it suffices to check only the extreme rays of the polyhedral cones which are the feasible sets of these 2^M linear programs.

Comments:
1. About Motivation. While checking the optimality on a non-differentiable point is a  mathematically interesting problem, it has little use in deep learning.  In practice, SGD often finds a global minimum easily of ReLU-activated deep neural networks [1].
2. This algorithm can only test if a point is a real SOSP. In practice, we can only hope to get an approximate SOSP. I expect a robust analysis, i.e., can we check whether it is a (\epsilon,\delta) SOSP?
3. About writing: g(z,\eta) and H(z,\eta) appear in Section 1 and Section 2, and they are used to define generalized SOSP. However, their formal definitions are in Lemma 2. I suggest give the formal definitions in Section 1 or Section 2 and give more intuitions on their formulas.

Minor Comments:
1. Many typos in references, e.g., cnn -> CNN.
2. Page 4: Big-Oh -> Big O.



Overall I think this paper presents some interesting ideas but I am unsatisfied with the issues above. I am happy to see the authors’ response, and I may modify my score.


[1] Zhang, C., Bengio, S., Hardt, M., Recht, B., & Vinyals, O. (2016). Understanding deep learning requires rethinking generalization. arXiv preprint arXiv:1611.03530.

---

> ### Author Response · Authors · 2018-11-22
> **Response to AnonReviewer4**
>
> We thank the reviewer for their efforts in reviewing our paper. We will address the concerns by the reviewer below:
>
> 1. Due to multiple reviewers raising a similar point, we addressed this issue in a separate comment above. Please refer to item (2) of the comment.
>
> 2. For the discussion on the robustness of the algorithm, we wrote a separate comment above to address common concerns raised by the reviewers. Please refer to item (1) of the comment.
>
> 3. Thank you for the suggestion! Since our analysis is specialized for ReLU neural networks, it would be a good idea to place Lemma 2 in Section 2.1. We will update the paper in our next revision.
>
> As for “minor comments”: Thank you for pointing out the typos. We will fix those issues as we revise our paper.

---

> > ### Comment · AnonReviewer4 · 2018-11-27
> > **Thanks for your response**
> >
> > Thanks for your response!
> > I have updated my review. I encourage author(s) to add the robust analysis and submit to the next top machine learning conference.

---

> > > ### Author Response · Authors · 2018-11-28
> > > **Response**
> > >
> > > Thank you for your response. Allow us to summarize our point again: we are providing the foundation for a practical check of local optimality that can be eventually turned into a useful algorithm in practice. Our paper and our review response express these results and contributions, acknowledging current limitations honestly and without any overclaims.
> > >
> > > In particular, our work is a theoretical contribution, and implementing (plus analyzing) a robust version requires much more additional effort, well beyond the current paper; to be fair, no non-trivial theory is developed in one day, and the amount of work that went into laying the foundations for a future robust analysis is fairly substantial already, in our opinion.
> > >
> > > We are disappointed by Reviewer 4’s decision to deduct their rating from 5 to 3. We believe that a rating of “clear rejection,” just because the paper does not build the whole tower already, is a bit harsh, as it ignores the importance of the foundations established herein. We would like to ask the reviewers to take this aspect into account in deciding their final ratings—ultimately, because we believe that the paper is well on-topic, and will spur follow up work.

---

> > > > ### Comment · AnonReviewer4 · 2018-11-29
> > > > **Clarification**
> > > >
> > > > Sorry for the confusion. What worries me most is not a practical implementation.
> > > >
> > > > From a theoretical point of view, the current version can only test if a point is a real SOSP. Thus this is only a qualitative result. I expect a theoretical machine learning paper in ICLR/ICML/NIPS/COLT to have at least some quantitative analysis on the error (even it has a large polynomial or exponential dependency), i.e., robust analysis for the problem studied in this paper.

---

### Author Response · Authors · 2018-11-22
**General remarks on commonly-raised concerns from reviewers**

Dear reviewers,

We truly appreciate the time and effort put in reviewing our paper, and we thank you all for your thoughtful comments and suggestions.

There were some concerns common to multiple reviewers, so we address them in a separate comment here.

(1) Regarding precision / robustness / ideal conditions issues
All reviewers raised concerns about numerical applicability of our algorithm. We would like to emphasize that this paper is a theoretical contribution seeking to understand the nondifferentiable points of the empirical risk surface of ReLU networks. As noted in the introduction, our understanding of nonsmooth points of empirical risk is limited, but in some cases, nonsmooth points can be precisely the “interesting points” that we care about. In this paper, we are theoretically/empirically showing that testing local optimality/second order stationarity at nondifferentiable points can be tractable (if the number of flat extreme rays is small), by exploiting the geometric structure of empirical risk.
But we fully agree (as also noted in Section 1 in the Remarks and in Section 5) that creating a numerically robust version of this algorithm that works for “close-to-nondifferential” points and approximate SOSPs will be needed before our theoretical work can attain its true practical significance---this goal requires a fairly substantial amount of effort (both theory and practice) and we hope to tackle it in the future.

(2) Regarding practical usefulness of the algorithm
Reviewers 1 and 4 raised concerns whether this algorithm is really meaningful in practice, given that SGD already performs well enough without our algorithm. It is true that in practice, SGD easily achieves near-zero empirical risk most of the time. However, please note that the solutions that we obtain at the end of training are not necessarily global or even local minima, because in practice we don’t have optimality tests / certificates during training.
In contrast, one of the most important beneficial features of convex optimization is existence of an optimality test (e.g., norm of the gradient is smaller than a certain threshold) for termination, which gives us a certificate of optimality. One of our motivations is that deep learning may also benefit from such optimality tests. Our analysis and experimental results suggest that even in the nonconvex and nonsmooth case of ReLU networks, it is sometimes not too difficult to get such a certificate of local optimality (we remind the readers that in general detecting local optimality for nonconvex problems is “NP-Hard”).
With a proper numerical implementation of our algorithm (although we leave it for future work), one can run a first-order method until it gets stuck near a point, and run our algorithm to test for optimality/second-order stationarity. If the point is an (approximate) SOSP, we can terminate without further computation time over many epochs; if the point has a descent direction, our algorithm will return a descent direction and we can continue on optimizing. Note that the descent direction may come from the second-order information; our algorithm even allows us to escape nonsmooth second-order saddle points.

We address the remaining points individually.

---

### Meta-Review · Area_Chair1 · 2018-12-12
**ICLR 2019 decision**

**Confidence:** 4
**Recommendation:** Accept (Poster)

**Metareview:**

This paper proposes a new method for verifying whether a given point of a two layer ReLU network is a local minima or a second order stationary point and checks for descent directions. All reviewers agree that the algorithm is based on number of new techniques involving both convex and non-convex QPs, and is novel. The method proposed in the paper has significant limitations as the method is not robust to handle approximate stationary points. Given these limitations, there is a disagreement between reviewers about the significance of the result . While I share the same concerns as R4, I agree with R3 and believe that the new ideas in the paper will inspire future work to extend the proposed method towards addressing these limitations. Hence I suggest acceptance.